# TwiBot-22: Towards Graph-Based Twitter Bot Detection

**Shangbin Feng**[1,2*] **Zhaoxuan Tan**[1*] **Herun Wan**[1*] **Ningnan Wang**[1*] **Zilong Chen**[1,3*] **Binchi Zhang**[1,4*]

**Qinghua Zheng**[1†] **Wenqian Zhang**[1] **Zhenyu Lei**[1] **Shujie Yang**[1] **Xinshun Feng**[1] **Qingyue Zhang**[1]

**Hongrui Wang**[1] **Yuhan Liu**[1] **Yuyang Bai**[1] **Heng Wang**[1] **Zijian Cai**[1] **Yanbo Wang**[1]

**Lijing Zheng**[1] **Zihan Ma**[1] **Jundong Li**[4] **Minnan Luo**[1]

Xi'an Jiaotong University[1], University of Washington[2], Tsinghua University[3], University of Virginia[4]

contact: `shangbin@cs.washington.edu`

## Abstract

Twitter bot detection has become an increasingly important task to combat misinformation, facilitate social media moderation, and preserve the integrity of the online discourse. State-of-the-art bot detection methods generally leverage the graph structure of the Twitter network, and they exhibit promising performance when confronting novel Twitter bots that traditional methods fail to detect. However, very few of the existing Twitter bot detection datasets are graph-based, and even these few graph-based datasets suffer from limited dataset scale, incomplete graph structure, as well as low annotation quality. In fact, the lack of a large-scale graph-based Twitter bot detection benchmark that addresses these issues has seriously hindered the development and evaluation of novel graph-based bot detection approaches. In this paper, we propose TwiBot-22, a comprehensive graph-based Twitter bot detection benchmark that presents the largest dataset to date, provides diversified entities and relations on the Twitter network, and has considerably better annotation quality than existing datasets. In addition, we re-implement 35 representative Twitter bot detection baselines and evaluate them on 9 datasets, including TwiBot-22, to promote a fair comparison of model performance and a holistic understanding of research progress. To facilitate further research, we consolidate all implemented codes and datasets into the TwiBot-22 evaluation framework, where researchers could consistently evaluate new models and datasets. The TwiBot-22 Twitter bot detection benchmark and evaluation framework are publicly available at `https://twibot22.github.io/`.

## 1 Introduction

Automated users on Twitter, also known as Twitter bots, have become a widely known and well-documented phenomenon. Over the past decade, malicious Twitter bots were responsible for a wide range of problems such as online disinformation [Cui et al., 2020, Wang et al., 2020, Lu and Li, 2020], election interference [Howard et al., 2016, Bradshaw et al., 2017, Rossi et al., 2020, Ferrara, 2017], extremism campaign [Ferrara et al., 2016, Marcellino et al., 2020], and even the spread of

---

[*] These authors contributed equally to this work.

[†] Corresponding author: Qinghua Zheng, School of Computer Science and Technology, Xi'an Jiaotong University. Email: qhzheng@xjtu.edu.cn; Minnan luo, School of Computer Science and Technology, Xi'an Jiaotong University. Email: minnluo@xjtu.edu.cn

36th Conference on Neural Information Processing Systems (NeurIPS 2022) Track on Datasets and Benchmarks.

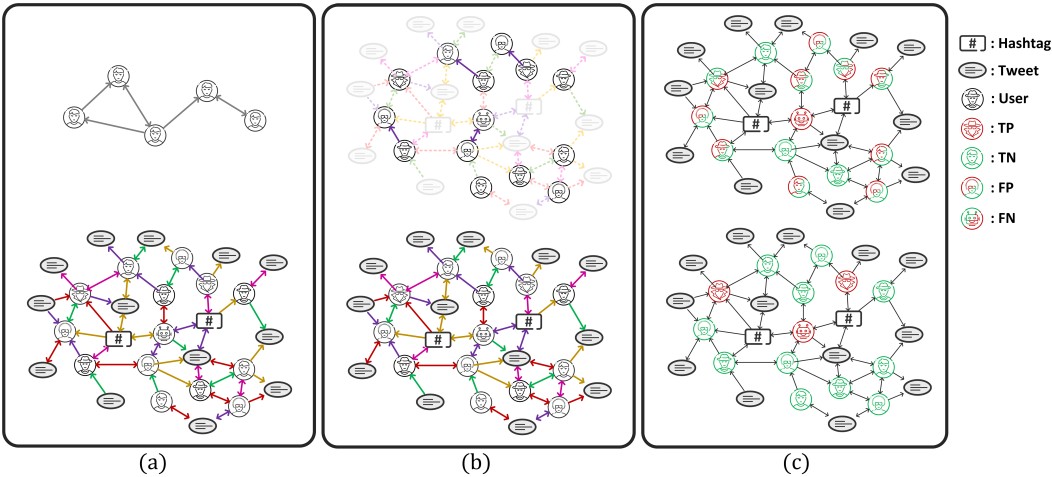

Figure 1: Compared to real-world Twitter (below), existing Twitter bot detection datasets (above) suffer from (a) limited dataset scale, (b) incomplete graph structure, and (c) poor annotation quality.

conspiracy theories [Ferrara, 2020, Ahmed et al., 2020, Anwar et al., 2021]. These societal challenges have called for automatic Twitter bot detection models to mitigate their negative influence.

Existing Twitter bot detection models are generally **feature-based**, where researchers propose to extract numerical features from user information such as metadata [Yang et al., 2020, Lee et al., 2011], user timeline [Mazza et al., 2019, Chavoshi et al., 2016], and follow relationships [Beskow and Carley, 2020, Chu et al., 2012]. However, feature-based approaches are susceptible to adversarial manipulation, *i.e.*, when bot operators try to avoid detection by tampering with these hand-crafted features [Cresci et al., 2017a, Cresci, 2020]. Researchers also proposed **text-based** approaches, where text analysis techniques such as word embeddings [Wei and Nguyen, 2019], recurrent neural networks [Kudugunta and Ferrara, 2018, Feng et al., 2021a], and pre-trained language models [Dukić et al., 2020] are leveraged to analyze tweet content and identify malicious intent. However, new generations of Twitter bots often intersperse malicious content with normal tweets stolen from genuine users [Cresci, 2020, Feng et al., 2021b], thus their bot nature becomes more subtle to text-based methods. With the advent of graph neural networks, recent advances focus on developing **graph-based** Twitter bot detection models. These methods [Ali Alhosseini et al., 2019, Feng et al., 2021b] interpret users as nodes and follow relationships as edges to leverage graph mining techniques such as GCN [Kipf and Welling, 2016], R-GCN [Schlichtkrull et al., 2018], and RGT [Feng et al., 2022] for graph-based bot detection. In fact, recent research have shown that graph-based approaches achieve state-of-the-art performance, are capable of detecting novel Twitter bots, and are better at addressing various challenges facing Twitter bot detection [Feng et al., 2021b, 2022].

However, the development and evaluation of graph-based Twitter bot detection models are poorly supported by existing datasets. The Bot Repository[3] provides a comprehensive collection of existing datasets. Out of the 18 listed datasets, only two of them, TwiBot-20 [Feng et al., 2021c] and cresci-2015 [Cresci et al., 2015], explicitly provide the graph structure among Twitter users. In addition, these two graph-based datasets suffer from the following issues as illustrated in Figure 1:

- **(a) limited dataset scale**. Twibot-20 [Feng et al., 2021c] contains 11,826 labeled users and cresci-15 [Cresci et al., 2015] contains 7,251 labeled users, while online conversations and discussions about heated topics often involve hundreds of thousands of users [Banda et al., 2021].

- **(b) incomplete graph structure**. Real-world Twitter is a heterogeneous information network that contains many types of entities and relations [Feng et al., 2022], while TwiBot-20 and cresci-15 only provide users and follow relationships.

- **(c) low annotation quality**. TwiBot-20 resorted to crowdsourcing for user annotation, while crowdsourcing leads to significant noise [Graells-Garrido and Baeza-Yates, 2022] and is susceptible to the false positive problem [Rauchfleisch and Kaiser, 2020].

---

[3]https://botometer.osome.iu.edu/bot-repository/datasets.html

In light of these challenges, we propose TwiBot-22, a graph-based Twitter bot detection benchmark that addresses these issues. Specifically, TwiBot-22 adopts a two-stage controlled expansion to sample the Twitter network, which results in a dataset that is 5 times the size of the largest existing dataset. TwiBot-22 provides 4 types of entities and 14 types of relations in the Twitter network, which provides the first (truly) heterogeneous graph for Twitter bot detection. Finally, TwiBot-22 adopts the weak supervision learning strategy for data annotation which results in significantly improved annotation quality. To compare TwiBot-22 with existing datasets, we re-implement 35 Twitter bot detection baselines and evaluate them on 9 datasets, including TwiBot-22, to provide a holistic view of research progress and highlight the advantages of TwiBot-22. We consolidate all datasets and implemented codes into the TwiBot-22 evaluation framework to facilitate further research. Our main contributions are summarized as follows:

- We propose TwiBot-22, a graph-based Twitter bot detection dataset that establishes the largest benchmark to date, provides diversified entities and relations in the Twitter network, and has considerably improved annotation quality.

- We re-implement and benchmark 35 existing Twitter bot detection models on 9 datasets, including TwiBot-22, to compare different approaches fairly and facilitate a holistic understanding of research progress in Twitter bot detection.

- We present the TwiBot-22 evaluation framework, where researchers could easily reproduce our results, examine existing datasets and methods, infer on unseen Twitter data, and contribute new datasets and models to the framework.

## 2 Related Work

### 2.1 Twitter Bot Detection

Existing Twitter bot detection methods mainly fall into three categories: **feature-based** methods, **text-based** methods, and **graph-based** methods.

**Feature-based** methods conduct feature engineering with user information and apply traditional classification algorithms for bot detection. Various features are extracted from user metadata [Kudugunta and Ferrara, 2018], tweets [Miller et al., 2014], description [Hayawi et al., 2022], temporal patterns [Mazza et al., 2019], and follow relationships [Feng et al., 2021a]. Later research efforts improve the scalability of feature-based approaches [Yang et al., 2020], automatically discover new bots [Chavoshi et al., 2016], and strike the balance between precision and recall [Morstatter et al., 2016]. However, as bot operators are increasingly aware of these hand-crafted features, novel bots often try to tamper with these features to evade detection [Cresci, 2020]. As a result, feature-based methods struggle to keep up with the arms race of bot evolution [Feng et al., 2021a].

**Text-based** methods use techniques in natural language processing to detect Twitter bots based on tweets and user descriptions. Word embeddings [Wei and Nguyen, 2019], recurrent neural networks [Kudugunta and Ferrara, 2018], the attention mechanism [Feng et al., 2021a], and pre-trained language models [Dukić et al., 2020] are adopted to encode tweets for bot detection. Later research combines tweet representations with user features [Cai et al., 2017], learns unsupervised user representations [Feng et al., 2021a], and attempts to address the multi-lingual issue in tweet content [Knauth, 2019]. However, novel bots begin to counter text-based approaches by diluting malicious tweets with content stolen from genuine users [Cresci, 2020]. In addition, Feng et al. [2021b] shows that analyzing tweet content alone might not be robust and accurate for bot detection.

**Graph-based** methods interpret Twitter as graphs and leverage concepts from network science and geometric deep learning for Twitter bot detection. Node centrality [Dehghan et al., 2022], node representation learning [Pham et al., 2022], graph neural networks (GNNs) [Ali Alhosseini et al., 2019], and heterogeneous GNNs [Feng et al., 2021b] are adopted to conduct graph-based Twitter bot detection. Later research try to combine graph-based and text-based methods [Guo et al., 2021a] or propose new GNN architectures to leverage heterogeneities in the Twitter network [Feng et al., 2022]. Graph-based approaches have shown great promise in tackling various challenges facing Twitter bot detection, such as bot communities and bot disguise [Feng et al., 2021b].

The development and evaluation of these models would not be possible without the many valuable Twitter bot detection datasets that were proposed over the past decade. These datasets mainly focus

on politics and elections in the United States [Yang et al., 2020] and European countries [Cresci et al., 2017a]. Cresci-17 [Cresci et al., 2017a] propose the concept of "social spambots" and presents a widely used dataset with more than one type of bots. TwiBot-20 [Feng et al., 2021c] is the latest and most comprehensive Twitter bot detection dataset that addresses the issue of user diversity in previous datasets. However, among 18 datasets presented in the Bot Repository, the go-to place for Twitter bot detection research, only 2 explicitly provide the graph structure of the Twitter network. In addition, these datasets suffer from limited dataset scale, incomplete graph structure, and low annotation quality while increasingly falling short of consistently benchmarking novel graph-based approaches. In light of these challenges, we present TwiBot-22 to alleviate these issues, promote a rethinking of research progress, and facilitate further research in Twitter bot detection.

## 2.2 Graph-based Social Network Analysis

Users in online social networks interact with each other and become part of the network structure, while the network structure is essential in understanding the patterns of social media [Carrington et al., 2005]. With the advent of geometric deep learning, graph neural networks (GNNs) have become increasingly popular in social network analysis research. Qian et al. [2021] propose to model social media with heterogeneous graphs and leverage relational GNNs for illicit drug trafficker detection. Guo et al. [2021b] propose dual graph enhanced embedding neural network to improve graph representation learning and tackle challenges in click-through rate prediction. Graphs and GNNs are also adopted to detect online fraud [Liu et al., 2021, Li et al., 2021, Wang et al., 2021, Mishra et al., 2021, Dou et al., 2020], combat misinformation [Cui et al., 2020, Wang et al., 2020, Lu and Li, 2020, Varlamis et al., 2022, Hu et al., 2021], and improve recommender systems [Ying et al., 2018, Wu et al.]. The task of Twitter bot detection is no exception, where novel and state-of-the-art approaches are increasingly graph-based [Ali Alhosseini et al., 2019, Magelinski et al., 2020, Feng et al., 2021b, 2022, Lei et al., 2022]. In this paper, we propose the TwiBot-22 benchmark to better support the development and evaluation of graph-based Twitter bot detection models.

## 3 TwiBot-22 Dataset

### 3.1 Data Collection

TwiBot-22 aims to present a large-scale and graph-based Twitter bot detection benchmark. To this end, we adopt a two-stage data collection process. We firstly adopt diversity-aware breadth-first search (BFS) to obtain the user network of TwiBot-22. We then collect additional entities and relations on the Twitter network to enrich the heterogeneity of the TwiBot-22 network.

**User network collection.** A common drawback of existing Twitter bot detection datasets is that they only feature a few types of bots and genuine users, while real-world Twitter is home to diversified users and bots [Feng et al., 2021c]. As a result, TwiBot-20 proposes to use breadth-first search (BFS) for user collection, starting from "seed users" and expanding with user follow relationships. To ensure that TwiBot-22 includes different types of bots and genuine users, we augment BFS with two diversity-aware sampling strategies:

- **Distribution diversity**. Given user metadata such as follower count, different types of users fall differently into the metadata distribution. Distribution diversity aims to sample users in the top, middle, and bottom of the distribution. For numerical metadata, among a user's neighbors and their metadata values, we select the $k$ users with the highest value, $k$ with the lowest, and $k$ randomly chosen from the rest. For true-or-false metadata, we select $k$ with true and $k$ with false.

- **Value diversity**. Given a user and its metadata, value diversity is adopted so that neighbors with significantly different metadata values are more likely to be included, ensuring the diversity of collected users. For numerical metadata, among expanding user $u$'s neighbors $X$ and their metadata values $x^{num}$, the probability of user $x \in X$ being sampled is denoted as $p(x) \propto |u^{num} - x^{num}|$. For true-or-false metadata we select $k$ users from the opposite class.

Based on these sampling strategies, TwiBot-22 conducts diversity-aware BFS starting from *@NeurIPSConf*. For each neighborhood expansion, one metadata and one of the sampling strategies are randomly adopted. The BFS process stops until the user network contains a desirable amount of Twitter users. More information about the user collection process is presented in Appendix A.2.

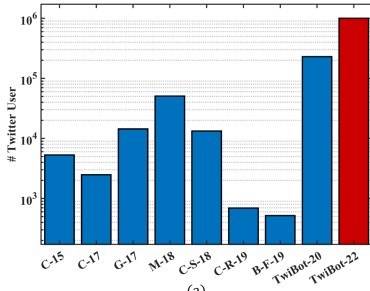

Figure 2: Analyzing TwiBot-22. (a) Dataset scale in terms of users. (b) List of entities and relations in TwiBot-22 and graph-based datasets. (c) Accuracy and F1-score of dataset labels compared to expert annotations as well as Randolph's kappa coefficient [Randolph, 2005] of expert agreement.

**Heterogeneous graph building.** The user network collection process constructs a homogeneous graph with users as nodes and follow relationships as edges. Apart from that, the Twitter network contains diversified entities and relations such as lists and retweets. Based on the user network, we collect the tweets, associated lists, and mentioned hashtags of these users as well as 12 other relations between users and these new entities. As a result, TwiBot-22 presents a heterogeneous Twitter network with 4 types of entities and 14 types of relations. More information about the heterogeneous Twitter network is presented in Appendix A.4.

As a result, we obtain the TwiBot-22 heterogeneous graph that contains 92,932,326 nodes and 170,185,937 edges. We present more dataset statistics in Table 7 in the appendix.

## 3.2 Data Annotation

Existing bot detection datasets often rely on manual annotation or crowdsourcing, while it is labor-intensive and thus not feasible with the large-scale TwiBot-22. As a result, we adopt weak supervision learning strategy to generate high-quality labels. We firstly invite bot detection experts to annotate 1,000 Twitter users in TwiBot-22. We then generate noisy labels with the help of bot detection models. Finally, we generate high-quality annotations for TwiBot-22 with the Snorkel framework [Ratner et al., 2017].

**Expert annotation.** We randomly select 1,000 users in TwiBot-22 and assign each user to 5 Twitter bot detection experts to identify if it is a bot. We then create an 8:2 split for these expert annotations as training and test sets. More details about these experts are presented in Appendix A.3.

**Generate noisy labels.** We employ 8 hand-crafted labeling functions and 7 competitive feature-based and neural network-based bot detection models to generate noisy labels. For handcrafted labeling functions, we adopt spam keywords in tweets and user descriptions as well as user categorical features such as verified and sensitive tweets. For feature engineering models, we select features based on users' metadata such as creation time, follower count and name length. We then adopt Adaboost, random forest, and MLP to result in three feature-based classifiers. For neural network-based models, we follow Feng et al. [2021b] to encode user information and employ MLP, GAT [Veličković et al., 2018], GCN [Kipf and Welling, 2016], and R-GCN [Schlichtkrull et al., 2018] as four classifiers. We train these classifiers on the training set of expert annotations and calculate the uncertainty scores for all users in TwiBot-22 under each classifier as $\phi = -(\hat{y}_0 \log(\hat{y}_0) + \hat{y}_1 \log(\hat{y}_1))$, where $\hat{y}_0$ and $\hat{y}_1$ denote the probability of being genuine users or bots. For each classifier, we then remove model predictions with the top 40% uncertainty scores to alleviate label noises.

**Majority voting.** After obtaining the noisy labels, we evaluate their plausibility with Snorkel [Ratner et al., 2017] and clean the labels at the same time. The output of the Snorkel system are probabilistic labels, thus we use these labels to train an MLP classifier to obtain the final annotations of TwiBot-22. We further evaluate the annotation quality on the test set of expert annotations and

Table 1: Statistics of the 9 datasets. TwiBot-20 contains unlabelled users so that # User $\neq$ # Human + # Bot. C-15, G-17, C-17, M-18, C-S-18, C-R-19, B-F-19 are short for cresci-2015, gilani-2017, cresci-2017, midterm-18, cresci-stock-2018, cresci-rtbust-2019, botometer-feedback-2019. C-17 contains only "post" edges between users and tweets, which is not a graph-based dataset.

| Dataset | C-15 | G-17 | C-17 | M-18 | C-S-18 | C-R-19 | B-F-19 | TwiBot-20 | TwiBot-22 |
|---|---|---|---|---|---|---|---|---|---|
| # Human | 1,950 | 1,394 | 3,474 | 8,092 | 6,174 | 340 | 380 | 5,237 | 860,057 |
| # Bot | 3,351 | 1,090 | 10,894 | 42,446 | 7,102 | 353 | 138 | 6,589 | 139,943 |
| # User | 5,301 | 2,484 | 14,368 | 50,538 | 13,276 | 693 | 518 | 229,580 | 1,000,000 |
| # Tweet | 2,827,757 | 0 | 6,637,615 | 0 | 0 | 0 | 0 | 33,488,192 | 88,217,457 |
| # Human Tweet | 2,631,730 | 0 | 2,839,361 | 0 | 0 | 0 | 0 | 33,488,192 | 81,250,102 |
| # Bot Tweet | 196,027 | 0 | 3,798,254 | 0 | 0 | 0 | 0 | 33,488,192 | 6,967,355 |
| # Edge | 7,086,134 | 0 | 6,637,615 | 0 | 0 | 0 | 0 | 33,716,171 | 170,185,937 |

we obtain an 90.5% accuracy. Compared to the 80% accuracy standard in TwiBot-20 [Feng et al., 2021c], TwiBot-22 has considerably imporved annotation quality.

## 3.3 Data Analysis

Existing graph-based Twitter bot detection datasets suffer from limited dataset scale, incomplete graph structure, and low annotation quality. As a result, we examine whether TwiBot-22 has adequately addressed these challenges and present our findings in Figure 2.

**Dataset scale.** Figure 2(a) illustrates the number of Twitter users in TwiBot-22 and existing datasets. It is illustrated that TwiBot-22 establishes the largest Twitter bot detection benchmark to date, with approximately 5 times more users than the second-largest TwiBot-20.

**Graph structure.** Figure 2(b) demonstrates that the TwiBot-22 network contains 4 types of entities and 14 types of relations, resulting in significantly enriched graph structure compared to existing graph-based datasets cresci-15 and TwiBot-20 with only 2 entity types and 3 relation types.

**Annotation quality.** TwiBot-20, the largest graph-based Twitter bot detection benchmark to date, leveraged crowdsourcing for data annotation. To improve label quality, TwiBot-22 uses weak supervision learning strategies and leverages 1,000 expert annotations to guide the process. To examine whether TwiBot-22 has improved annotation quality than TwiBot-20, we ask Twitter bot detection experts to participate in an "expert study", where they are asked to evaluate users in TwiBot-20 and TwiBot-22 to examine how often do experts agree with dataset labels. Figure 2(c) illustrates the results, which shows that these experts find TwiBot-22 to provide more consistent, accurate, and trustworthy data annotations. More details about the expert study are available in Appendix A.5.

## 4 Experiments

### 4.1 Experiment Settings

**Datasets.** We evaluate Twitter bot detection models on all 9 datasets in the Bot Reporistory that contain both bots and genuine users: cresci-2015 [Cresci et al., 2015], gilani-2017 [Gilani et al., 2017], cresci-2017 [Cresci et al., 2017a,b], midterm-18 [Yang et al., 2020], cresci-stock-2018 [Cresci et al., 2018, 2019], cresci-rtbust-2019 [Mazza et al., 2019], botometer-feedback-2019 [Yang et al., 2019], TwiBot-20 [Feng et al., 2021c], and TwiBot-22. We present dataset details in Table 1. We create a 7:2:1 random split as training, validation, and test set to ensure fair comparison.

**Baselines.** We re-implement and evaluate 35 Twitter bot detection baselines SGBot [Yang et al., 2020], Kudugunta and Ferrara [2018], Hayawi et al. [2022], BotHunter [Beskow and Carley, 2018], NameBot [Beskow and Carley, 2019], Abreu et al. [2020], Cresci et al. [2016], Wei and Nguyen [2019], BGSRD [Guo et al., 2021a], RoBERTa [Liu et al., 2019], T5 [Raffel et al., 2020], Efthimion et al. [2018], Kantepe and Ganiz [2017], Miller et al. [2014], Varol et al. [2017], Kouvela et al. [2020], Ferreira Dos Santos et al. [2019], Lee et al. [2011], LOBO [Echeverrï£¡ a et al., 2018],

Table 2: Average bot detection accuracy and standard deviation of five runs of 35 baseline methods on 9 datasets. **Bold** and underline indicate the highest and second highest performance. The F, T, and G in the "Type" column indicates whether a baseline is feature-based, text-based, or graph-based. Cresci *et al.* and Botometer are deterministic methods or APIs without standard deviation. "/" indicates that the dataset does not contain enough user information to support the baseline. "-" indicates that the baseline is not scalable to the largest TwiBot-22 dataset.

| Method | Type | C-15 | G-17 | C-17 | M-18 | C-S-18 | C-R-19 | B-F-19 | TwiBot-20 | TwiBot-22 |
|---|---|---|---|---|---|---|---|---|---|---|
| SGBot | F | 77.1 (±0.2) | **78.6** (±0.8) | 92.1 (±0.3) | 99.2 (±0.0) | 81.3 (±0.1) | 80.9 (±1.5) | 75.5 (±1.9) | 81.6 (±0.5) | 75.1 (±0.1) |
| Kudugunta *et al.* | F | 75.3 (±0.1) | 70.0 (±1.1) | 88.3 (±0.2) | 91.0 (±0.5) | 77.5 (±0.1) | 62.9 (±0.8) | 74.0 (±4.7) | 59.6 (±0.7) | 65.9 (±0.0) |
| Hayawi *et al.* | F | 84.3 (±0.0) | 52.7 (±0.0) | 90.8 (±0.0) | 84.6 (±0.0) | 50.0 (±0.0) | 51.2 (±0.0) | 77.0 (±0.0) | 73.1 (±0.0) | 76.5 (±0.0) |
| BotHunter | F | 96.5 (±1.2) | 76.4 (±1.0) | 88.1 (±0.2) | **99.3** (±0.0) | 81.2 (±0.2) | 81.5 (±1.7) | 74.7 (±1.0) | 75.2 (±0.4) | 72.8 (±0.0) |
| NameBot | F | 77.0 (±0.0) | 60.8 (±0.0) | 76.8 (±0.0) | 85.1 (±0.0) | 55.8 (±0.0) | 63.2 (±0.0) | 71.7 (±0.0) | 59.1 (±0.1) | 70.6 (±0.0) |
| Abreu *et al.* | F | 75.7 (±0.1) | 74.3 (±0.1) | 92.7 (±0.1) | 96.5 (±0.1) | 75.4 (±0.1) | 80.9 (±0.1) | **77.4** (±0.1) | 73.4 (±0.1) | 70.7 (±0.1) |
| Cresci *et al.* | T | 37.0 | / | 33.5 | / | / | / | / | 47.8 | - |
| Wei *et al.* | T | 96.1 (±1.4) | / | 89.3 (±0.7) | / | / | / | / | 71.3 (±1.6) | 70.2 (±1.2) |
| BGSRD | T | 87.8 (±0.6) | 48.5 (±8.4) | 75.9 (±0.0) | 82.9 (±1.5) | 50.7 (±1.3) | 50.0 (±4.9) | 59.6 (±3.1) | 66.4 (±1.0) | 71.9 (±1.8) |
| RoBERTa | T | 97.0 (±0.1) | / | 97.2 (±0.0) | / | / | / | / | 75.5 (±0.1) | 72.1 (±0.1) |
| T5 | T | 92.3 (±0.1) | / | 96.4 (±0.0) | / | / | / | / | 73.5 (±0.1) | 72.1 (±0.1) |
| Efthimion *et al.* | FT | 92.5 (±0.0) | 55.5 (±0.0) | 88.0 (±0.0) | 93.4 (±0.0) | 70.8 (±0.0) | 67.6 (±0.0) | 69.8 (±0.0) | 62.8 (±0.0) | 74.1 (±0.0) |
| Kantepe *et al.* | FT | 97.5 (±1.3) | / | 98.2 (±1.5) | / | / | / | / | 80.3 (±4.3) | 76.4 (±2.4) |
| Miller *et al.* | FT | 75.5 (±0.0) | 51.0 (±0.0) | 77.1 (±0.2) | 83.7 (±0.0) | 52.5 (±0.0) | 54.4 (±0.0) | **77.4** (±0.0) | 64.5 (±0.4) | 30.4 (±0.1) |
| Varol *et al.* | FT | 93.2 (±0.5) | / | / | / | / | / | / | 78.7 (±0.6) | 73.9 (±0.0) |
| Kouvela *et al.* | FT | 97.8 (±0.5) | 74.7 (±0.9) | 98.4 (±0.1) | 97.0 (±0.1) | 79.3 (±0.3) | 79.7 (±1.2) | 71.3 (±0.9) | 84.0 (±0.4) | 76.4 (±0.0) |
| Santos *et al.* | FT | 70.8 (±0.0) | 51.4 (±0.0) | 73.8 (±0.0) | 86.6 (±0.0) | 62.5 (±0.0) | 73.5 (±0.0) | 71.7 (±0.0) | 58.7 (±0.0) | - |
| Lee *et al.* | FT | 98.2 (±0.1) | 74.8 (±1.2) | **98.8** (±0.1) | 96.4 (±0.1) | 81.5 (±0.4) | **83.5** (±1.9) | 75.5 (±1.3) | 77.4 (±0.5) | 76.3 (±0.1) |
| LOBO | FT | **98.4** (±0.3) | / | 96.6 (±0.3) | / | / | / | / | 77.4 (±0.2) | 75.7 (±0.1) |
| Moghaddam *et al.* | FG | 73.6 (±0.2) | / | / | / | / | / | / | 74.0 (±0.8) | 73.8 (±0.0) |
| Alhosseini *et al.* | FG | 89.6 (±0.6) | / | / | / | / | / | / | 59.9 (±0.6) | 47.7 (±8.7) |
| Knauth *et al.* | FTG | 85.9 (±0.0) | 49.6 (±0.0) | 90.2 (±0.0) | 83.9 (±0.0) | **88.7** (±0.0) | 50.0 (±0.0) | 76.0 (±0.0) | 81.9 (±0.0) | 71.3 (±0.0) |
| FriendBot | FTG | 96.9 (±1.1) | / | 78.0 (±1.0) | / | / | / | / | 75.9 (±0.5) | - |
| SATAR | FTG | 93.4 (±0.5) | / | / | / | / | / | / | 84.0 (±0.8) | - |
| Botometer | FTG | 57.9 | 71.6 | 94.2 | 89.5 | 72.6 | 69.2 | 50.0 | 53.1 | 49.9 |
| Rodrifuez-Ruiz *et al.* | FTG | 82.4 (±0.0) | / | 76.4 (±0.0) | / | / | / | / | 66.0 (±0.1) | 49.4 (±0.0) |
| GraphHist | FTG | 77.4 (±0.2) | / | / | / | / | / | / | 51.3 (±0.3) | - |
| EvolveBot | FTG | 92.2 (±1.7) | / | / | / | / | / | / | 65.8 (±0.6) | 71.1 (±0.1) |
| Dehghan *et al.* | FTG | 62.1 (±0.0) | / | / | / | / | / | / | 86.7 (±0.1) | - |
| GCN | FTG | 96.4 (±0.0) | / | / | / | / | / | / | 77.5 (±0.0) | 78.4 (±0.0) |
| GAT | FTG | 96.9 (±0.2) | / | / | / | / | / | / | 83.3 (±0.0) | 79.5 (±0.0) |
| HGT | FTG | 96.0 (±0.3) | / | / | / | / | / | / | **86.9** (±0.2) | 74.9 (±0.1) |
| SimpleHGN | FTG | 96.7 (±0.5) | / | / | / | / | / | / | 86.7 (±0.2) | 76.7 (±0.3) |
| BotRGCN | FTG | 96.5 (±0.7) | / | / | / | / | / | / | 85.8 (±0.7) | **79.7** (±0.1) |
| RGT | FTG | 97.2 (±0.3) | / | / | / | / | / | / | 86.6 (±0.4) | 76.5 (±0.4) |

Moghaddam and Abbaspour [2022], Ali Alhosseini et al. [2019], Knauth [2019], FriendBot [Beskow and Carley, 2020], SATAR [Feng et al., 2021a], Botometer [Yang et al., 2022], Rodríguez-Ruiz et al. [2020], GraphHist [Magelinski et al., 2020], EvolveBot [Yang et al., 2013], Dehghan et al. [2022], GCN [Kipf and Welling, 2016], GAT [Veličković et al., 2018], HGT [Hu et al., 2020], SimpleHGN [Lv et al., 2021], BotRGCN [Feng et al., 2021b], RGT [Feng et al., 2022]. These methods leverage different aspects of user information and represent different stages of bot detection research. More details about these baseline methods are available in Appendix B.1.

## 4.2 Experiment Results

We re-implement 35 baseline methods and evaluate them on 9 Twitter bot detection datasets. We run each baseline method for **five times** and report the average performance and standard deviation. Table 2 presents the benchmarking results. Our main discoveries are summarized as follows:

- Graph-based approaches are generally more effective than feature-based or text-based methods. As a matter of fact, all top 5 models on TwiBot-20 and TwiBot-22 are graph-based. On average, these top-5 graph-based methods outperform the global average of all baselines by 13.8% and 8.2% on TwiBot-20 and TwiBot-22. This trend suggests that future research in Twitter bot detection should further examine how users and bots interact on Twitter and the heterogeneous graph structure thus formed.

Table 3: Removing the graph-related model component from graph-based methods (w/o G) while comparing to their original versions (Prev.) on TwiBot-20 and TwiBot-22.

| Method | TwiBot-20 | | | | | | TwiBot-22 | | | | | |
| --- | --- | --- | --- | --- | --- | --- | --- | --- | --- | --- | --- | --- |
| | Acc | | | F1 | | | Acc | | | F1 | | |
| | Prev. | w/o G | Diff. | Prev. | w/o G | Diff. | Prev. | w/o G | Diff. | Prev. | w/o G | Diff. |
| Ali Alhosseini et al. [2019] | 59.9 | 61.8 | +1.9 | 72.1 | 70.7 | −1.4 | 70.7 | 66.9 | −3.8 | 5.7 | 3.5 | −2.2 |
| Moghaddam and Abbaspour [2022] | 74.0 | 72.2 | −1.8 | 77.9 | 75.8 | −2.1 | 73.8 | 73.3 | −0.5 | 32.1 | 32.0 | −0.1 |
| Knauth [2019] | 81.9 | 81.4 | −0.5 | 85.2 | 84.9 | −0.3 | 71.3 | 71.5 | +0.2 | 37.1 | 11.3 | −25.8 |
| EvolveBot [Yang et al., 2013] | 65.8 | 65.1 | −0.7 | 69.7 | 69.3 | −0.4 | 71.1 | 71.0 | −0.1 | 14.1 | 14.0 | −0.1 |
| BotRGCN [Feng et al., 2021b] | 85.7 | 82.6 | −3.1 | 87.3 | 83.8 | −3.5 | 79.7 | 75.4 | −4.3 | 57.5 | 41.2 | −16.3 |
| RGT [Feng et al., 2022] | 86.6 | 82.6 | −4.0 | 88.0 | 83.8 | −4.2 | 76.5 | 75.4 | −1.1 | 42.9 | 41.2 | −1.7 |

- Most existing datasets do not provide the graph structure of Twitter users to support graph-based approaches, while TwiBot-22 supports all baseline methods and serve as a comprehensive evaluation benchmark. As novel and state-of-the-art models are increasingly graph-based, future Twitter bot detection datasets should provide the graph structure of real-world Twitter.

- TwiBot-22 establishes the largest benchmark while exposing the scalability issues of baseline methods. For example, Dehghan et al. [2022] achieves near-sota performance on TwiBot-20, while failing to scale to TwiBot-22 as our implementation encounters the out-of-memory problem.

- Performance on TwiBot-22 is on average 2.7% lower than on TwiBot-20 across all baseline methods, which demonstrates that Twitter bot detection is still an open problem that calls for further research. This could be attributed to the fact that Twitter bots are constantly evolving to improve their disguise and evade detection, thus bot detection methods should also adapt and evolve.

### 4.3 Removing Graphs from Baselines

Benchmarking results in Table 2 demonstrate that graph-based approaches generally achieve better performance. To examine the role of graphs in graph-based approaches, we remove the graph component in competitive graph-based methods [Ali Alhosseini et al., 2019, Moghaddam and Abbaspour, 2022, Knauth, 2019, Yang et al., 2013, Feng et al., 2021b, 2022] and report model performance in Table 3. It is demonstrated that:

- All baseline methods exhibit performance drops to different extents on two datasets when the graph component is removed. This indicates that the graph-related components in graph-based approaches contribute to bot detection performance.

- For graph neural network-based approaches BotRGCN [Feng et al., 2021b] and RGT [Feng et al., 2022], the performance drop is generally more severe. This suggests that graph neural networks play an important role in boosting model performance and advancing bot detection research.

More details about how graphs are removed from baseline methods are presented in Appendix B.4.

### 4.4 Generalization Study

The challenge of generalization [Yang et al., 2020, Feng et al., 2021a], *i.e.*, whether Twitter bot detection models perform well on unseen data, is essential in ensuring that bot detection research translates to effective social media moderation and real-world impact. To evaluate the generalization ability of existing Twitter bot detection approaches, we identify 10 sub-communities in the TwiBot-22 network. We then use these sub-communities as folds and examine the performance of several representative models when trained on fold $i$ and evaluated on fold $j$. Figure 3 illustrates that:

- **Graph-based methods are better at generalizing to unseen data.** For example, BotRGCN [Feng et al., 2021b] achieves the best avg score among all baseline methods, outperforming the second-highest RGT by 3.66. This suggests that leveraging the network structure of Twitter might be a potential solution to the generalization challenge.

- **Good model performance does not necessarily translate to good generalization ability.** For example, GAT outperforms LOBO by 5.9% and 3.8% on TwiBot-20 and TwiBot-22 respectively

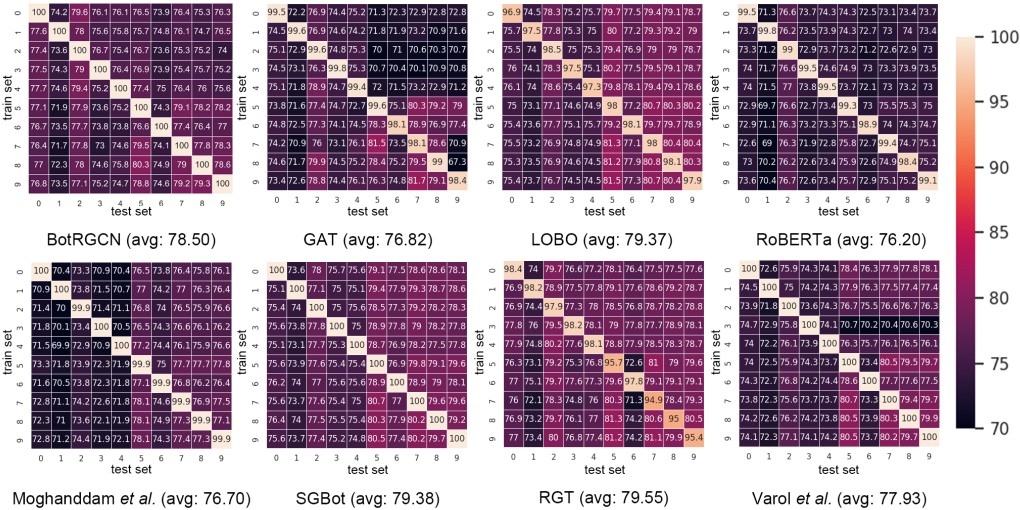

Figure 3: Training models on fold $i$ and testing on fold $j$. We present model accuracy and report the average value of each heatmap (avg), which serves as an overall indicator of generalization ability.

in terms of accuracy. However, GAT has lower avg (-2.55) compared to LOBO. This suggests that future bot detection research should focus on generalization in addition to model performance.

More details about the 10 sub-communities are provided in Appendix B.5.

# 5   Evaluation Framework

We consolidate Twitter bot detection datasets, data prepossessing codes, and all 35 implemented baselines into the TwiBot-22 evaluation framework and make it publicly available. We hope our efforts would facilitate further research in Twitter bot detection through:

- establishing a unified interface for different types of Twitter bot detection datasets
- providing 35 representative baselines and well-documented implementations
- enriching the evaluation framework with new datasets and methods proposed in future research

Please refer to `https://twibot22.github.io/` for more details.

# 6   Conclusion and Future Work

In this paper, we propose TwiBot-22, a graph-based Twitter bot detection benchmark. TwiBot-22 successfully alleviates the challenges of limited dataset scale, incomplete graph structure, and poor annotation quality in existing datasets. Specifically, we employ a two-stage data collection process and adopt the weak supervision learning strategy for data annotation. We then re-implement 35 representative Twitter bot detection models and evaluate them on 9 datasets, including TwiBot-22, to promote a holistic understanding of research progress. We further examine the role of graphs in graph-based methods and the generalization ability of competitive bot detection baselines. Finally, we consolidate all implemented codes into the TwiBot-22 evaluation framework, where researchers could easily reproduce our experiments and quickly test out new datasets and models.

Armed with the TwiBot-22 benchmark and the TwiBot-22 evaluation framework, we aim to investigate these research questions in the future:

- **How do we identify bot clusters and their coordination campaigns?** While existing works study Twitter bot detection through individual analysis, novel Twitter bots are increasingly observed to act in groups and launch coordinated attacks. We aim to complement the scarce literature by proposing temporal and subgraph-level bot detection approaches to address this issue.

- **How do we incorporate multi-modal user features for bot detection?** In addition to text and graph, Twitter users and bots generate multi-modal user information such as images and videos. Since TwiBot-22 provides user media while none of the 35 baselines leverage these modalities, we aim to further explore Twitter bot detection with the help of images and videos.

- **How do we evaluate the generalization ability of bot detection methods?** Existing works mainly focus on bot detection performance while generalization is essential in ensuring that bot detection research generates real-world impact. We aim to complement the scarce literature by proposing measures to quantitatively evaluate bot detection generalization.

- **How do we improve the scalability of graph-based models?** Existing graph-based bot detection methods demand significantly more computation resources and execution time than feature-based models. Given that the Twitter network is rapidly expanding, we aim to further explore scalable and graph-based bot detection methods.

## Acknowledgements

This work was supported by the National Key Research and Development Program of China (No. 2020AAA0108800), National Nature Science Foundation of China (No. 62192781, No. 62272374, No. 61872287, No. 62250009, No. 62137002), Innovative Research Group of the National Natural Science Foundation of China (61721002), Innovation Research Team of Ministry of Education (IRT_17R86), Project of China Knowledge Center for Engineering Science and Technology and Project of Chinese academy of engineering "The Online and Offline Mixed Educational Service System for The Belt and Road Training in MOOC China".

We would like to thank the reviewers and area chair for the constructive feedback. We would also like to thank all LUD lab members for our collaborative research environment and for making our bot detection research series possible. As Shangbin concludes his term as the director and Zhaoxuan begins his term, we hope the LUD lab will continue to thrive in the years to come.

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
