# OpenReview forum: "TwiBot-22: Towards Graph-Based Twitter Bot Detection"
_NeurIPS.cc/2022/Track/Datasets_and_Benchmarks — NeurIPS 2022 Datasets and Benchmarks _

### Official Review · Reviewer_Mgrm · 2022-07-19
**Some meaningful contributions but also some issues**

**Rating:** 7
**Confidence:** 4
**Correctness:** 1. In the main benchmarking results i…

**Strengths:**

1. The new dataset contributed is significantly larger (more users) and more comprehensive (more data types/graph relations) than previous bot detection datasets.

2. The benchmarking is extensive, with 35 models over 9 datasets.

3. The dataset and benchmarking codes are accessible.

This is an important real-world problem, with both social good and business applications. Given that and the points above, it seems highly likely that this work will be useful for future researchers.


**Weaknesses:**

1. I have doubts that accuracy is an appropriate metric for the benchmark with the proposed dataset, due to class imbalance. This may affect reliability of some of the conclusions - at the least, I feel they should be justified with another metric. I elaborate in detail below (point #1 under Correctness).

2. Because the annotation process involves several specific model architectures, I am concerned the benchmarking might have some bias in favor of similar models (point #2 under Correctness).

3. There are a few parts which are not completely clear to me. Most importantly, some of the expert annotation process and corresponding evaluation of the dataset annotations (point #4 and part of point #5 under Clarity).



**Additional Feedback:**

1. Two expert hand-labeled subsets of users were collected during the annotation process. It might be interesting to see results of the different benchmarked models on these subsets, since in theory they are the most accurate labels available. Also, are they available or marked somewhere in the dataset (on github/drive/etc.)? I didn't find them, but only looked briefly.

2. The Snorkel model used for annotation seems like it might have better performance than any benchmarked model, at least according to the first set expert labels where it achieved 90.5% accuracy. Not crucial to the paper, but besides annotation, is it worth considering that as a future method? Would it perform well on the other datasets benchmarked here?

3. Overall, I feel this paper makes some clear and worthwhile contributions. For example, I fully agree with the authors that larger and more comprehensive data is important for this task, and the dataset contributed will surely be useful for future work on bot detection. However, as explained above, I have some concerns about the validity of the benchmarking results (chief among them the choice of accuracy as metric). There are also some sections where additional details would improve the clarity, not just in terms of ease of reading but for properly understanding and using the dataset and results. Therefore, I currently recommend reject, but would welcome rebuttal and/or revisions addressing these points.

Regardless of the final decision here,  the author's contributions to this important area and hope they will continue research along these lines. Thank you!


EDIT: the authors have done substantial revisions and new experiments that significantly improve the quality of the paper and address most of my concerns. I am raising my overall score accordingly (previously 5 now 7).

**Clarity:**

The paper is generally smooth to read. There are a few sections I have questions or concerns about:

1. For the expansion with k users discussed on lines 151-160, I couldn't find any explicit value of k. The paper does discuss adding 6 users at a time in appendix A.2, so my best guess is k=2. But from lines 154-155, it seems that there should be k*2 users added for a true-false metadata but k*3 for others, i.e. different values added depending on whether true-false or not. I suggest in the appendix explicitly saying the value of k, and if indeed different numbers of users are added depending on true-false or not, saying the two total numbers added per step (e.g. "we add 6 users per step, except for true-false metadata where we add 4"). If it's always 6 users regardless of true-false or not, then I suggest making k the total number of users added, and rewriting 151-160 in terms of fraction of k (e.g. k/3 for each of highest/lowest/random on line 154, k/2 for true-false on line 155, etc.).

2. When collecting additional tweets related to a hashtag, e.g. lines 597-598, does this include all tweets that include that hashtag during the collection period? Or only some subset, e.g. a fixed amount per hashtag?

3. In the generalization study, it would be helpful to elaborate further (e.g. in appendix) on exactly how the subcommunities are constructed. For the user-based ones, are they the one-hop neighborhoods (based on follow relation?) of the 5 selected users? For the hashtag clustering ones, how is word2vec applied, and is anything special done to make it work with hashtags (which are often not dictionary words)? When applying K-means to those representations, is K=5, or else what is the value and how was it chosen? Is a user included in the sub-community as long as they use a hashtag within it, or is there a stricter criterion?

4. On line 199 the labels achieve 90.5% accuracy compared to the expert hand-labeled test set. But then in figure 2c the paper reports accuracy against a different expert hand-labeled set and it's a bit under 80%. This difference seems somewhat large to be random? Do the authors have any comments, explanation, or hypothesis here?
Possibly related, it is not clear to me how the 5-way labels the experts provide (in the second case) are converted to the 2-way labels in the dataset to make this comparison. How are disagreements resolved between the 3 experts providing a label for each example? Similar question for the 1000 expert annotations used in the dataset construction itself. The paper explains each example is labeled by 5 experts as human/bot/not sure, and then majority voting is applied, but how are cases without a majority (e.g. 2x human 2x not sure 1x bot) resolved? What happens to cases with majority not sure, are they dropped, and if so is the 1000 the remainder after dropping such cases or is the actual number used somewhat less than 1000?

5. In Figure 2c, is the F1 reported for the positive class "bot" or something else e.g. macro-average? I would guess that the results in Table 8 are F1 for the positive class "bot" but would also be good to confirm.

There are a few typos such as:

Line 163: "stops until the user network contains..." -> should read "stops when the ..." or "continues until the ..."

Line 200: imporved -> improved

Line 221: bot reporistory -> bot repository

Line 599: period missing at end of sentence




**Documentation:**

Yes, the paper has documentation and there is also a github website and repo with the dataset, benchmarking code, and related information.

**Ethics:**

The paper notes "We need to make sure that the TwiBot-22 dataset and evaluation framework should not be abused to design advanced Twitter bots" (lines 649-650) but does not elaborate. I do feel this research is likely to have net positive effect here (i.e. helping more for detecting bots than designing more advanced bots), don't see this as a serious flaw in the paper, and appreciate that the authors acknowledged it. The statement could be strengthened though with any steps taken, recommended, or just considered to avoid negative outcomes. For example, a number of datasets require a brief application form for access; could something like that help ensure this data is used for academic research purposes, or would that limit accessibility more than it is worth?

**Relation To Prior Work:**

Generally yes. The paper discusses the data types different methods use, and summarizes how the benchmarked methods work. It discusses previous datasets for this task and how the new dataset presented compares. I think the literature review could be improved by adding discussion of performance metrics used in the bot detection literature (please see point #1 under Correctness for elaboration).

**Summary And Contributions:**

The paper provides a new, larger and more comprehensive dataset for Twitter bot detection. It benchmarks a large number of existing methods on the new dataset and 8 others from the literature. There are also some experiments on the importance of graphs in several models and on how well different models generalize.

---

> ### Author Response · Authors · 2022-08-10
> **Evaluation metrics for Twitter bot detection. (1/14)**
>
> Thank you for your detailed and thoughtful review of our submission: *TwiBot-22: Towards Graph-Based Twitter Bot Detection*. We hereby address the raised questions and concerns. We believe the paper would be much stronger thanks to your feedback.
>
> ```
> "While even F1 as a metric for binary classification is subject to debate, to my knowledge it is at least much better than accuracy when there is substantial class imbalance. I suggest at least replacing Table 2 with Table 8. Then in appendix present precision and recall individually to add more detail. It would be even better if the authors could add some discussion on the choice of metric. For example, what is standard in the literature? If the authors agree that accuracy is perhaps not appropriate here, is F1 justified or is there something better?"
> ```
>
> We surveyed all baseline methods adopted in this paper. Out of the 29 baseline methods that are specific to the task of bot detection, 22 used accuracy, 20 used F1-score, 16 used recall, 15 used AUC, and 14 used precision. As a result, we conclude that accuracy and F1-score are the most prevalent metrics in previous bot detection literature.
>
> We agree that for an imbalanced dataset such as TwiBot-20, F1-score might be more important than accuracy and more metrics could be reported. In addition, we hereby present precision and recall results. We will add these results to the paper to facilitate holistic understanding.
>
> https://github.com/LuoUndergradXJTU/TwiBot-22#precision
>
> https://github.com/LuoUndergradXJTU/TwiBot-22#recall

---

> > ### Comment · Reviewer_Mgrm · 2022-08-17
> > **Re: 1/14**
> >
> > Thank you for the detailed responses. Following the numbering in the individual comments, I feel 5, 6, and 9-13 are fully addressed between the comments and the changes made or planned for the paper and supplementary material. I have some followup questions/comments, or parts I don't feel are clear yet, on others - I reply to those individually.
> >
> > Overall, I feel the authors have addressed many of my concerns. I will continue any discussion here (time permitting) and read the rest of the discussion on other reviews before I make any changes, but I anticipate raising my overall score.
> >
> > Regarding metrics, thank you for surveying the baseline methods' metrics and providing precision and recall; I think that will help evaluating and understanding the methods' performance.

---

> > > ### Author Response · Authors · 2022-08-22
> > > **Thank you very much.**
> > >
> > > Thank you for your continuing feedback. You successfully highlighted many issues where attention is needed and I am glad that TwiBot-22 will be much stronger thanks to your feedback. It is nice having you as one of our reviewers.
> > >
> > > There is one more week in the discussion phase and let me know if you have more suggestions. In the meantime, I wonder if it would be possible for you to kindly revisit your overall score. That would be very helpful. :)

---

> > > > ### Comment · Reviewer_Mgrm · 2022-08-26
> > > > **Revisiting Overall Score**
> > > >
> > > > Sure, thank you for the meaningful research.
> > > >
> > > > I've considered the discussion and changes overall, and I feel the revisions/additional explanations/new experiments improve the paper substantially. I've raised my overall score from 5 to 7.

---

> ### Author Response · Authors · 2022-08-10
> **Updated experiment conclusions. (2/14)**
>
> ```
> "I think some of the conclusions would also need to be revised accordingly. Of the 4 conclusions lines 244-259, the second and third do not depend on the metric so would remain unchanged. Assuming we go by F1, the first would need revised numbers, but just eyeballing it perhaps the overall result that graph-based methods perform better on average still holds? The fourth, regarding performance drop between TwiBot-20 and TwiBot-22, seems much more extreme. Many of the models seem to decay by 30+ percentage points, with some even dropping to near 0. Do the authors have any hypotheses on the cause? Perhaps these models can't handle this extent of class imbalance? Have the authors done any experiments balancing the training data, for example sampling an equal number of human and bot users to train on?"
> ```
>
> In light of new metrics and new results, we will revise the conclusions accordingly. Now 5 and 4 of the top-5 methods are graph-based on TwiBot-20 and TwiBot-22. In addition, graph-based methods outperform the global average by 6.99% and 3.79% on TwiBot-20 and TwiBot-22.
>
> There were some typos that resulted in near-0 F1-scores and huge performance decay in Table 8. We apologize for these mistakes and correct them here:
>
> https://github.com/LuoUndergradXJTU/TwiBot-22#f1
>
> The reviewer raised an important point about data imbalance in TwiBot-22. We did not employ data balancing techniques in the experiments, aiming to simulate the real-world scenario where bot:user is not 1:1 and test out different baselines’ ability under an imbalanced dataset. As a result, methods that are more robust to dataset imbalance exhibit better F1-score on TwiBot-22.

---

> > ### Comment · Reviewer_Mgrm · 2022-08-17
> > **Re: 2/14**
> >
> > Thank you for the updated table.
> >
> > If I've done the math correctly, based on approximately 14% positive examples, predicting always positive will give about 24.5 F1. It seems 5 methods are worse than this naive baseline, and several more only marginally better. Considering most of these methods performed significantly better than a naive baseline on TwiBot-20, if the difference is solely due to more sophisticated bots, it would seem that in two years the bots not just became harder for these methods to detect but nearly impossible. Getting a bit speculative here, but I'm somewhat skeptical of this. Reason being, presumably these bots are from a variety of sources and purposes, so it feels unlikely to me that near 100% of bots with the old behavior that those models worked on have been completely phased out in two years. If that were really the case, it seems like a pretty interesting topic for future research (what exactly was changed that makes those models fail? How do not just e.g. sophisticated state actors but also more mundane bots evolve so fast?).
> >
> > I would be less surprised though if a large part of the difference is instead due to changes in the data collection rather than exclusively bots themselves. Such as the imbalance - it seems like most previous datasets, judging from the table in the paper, were more balanced. Other factors like the new labeling process here could also play a role.
> >
> > Regarding imbalanced data, I agree that the approach of the paper - NOT balancing it - is more realistic for evaluation. However, I wonder if some methods would benefit from it in training only. For example, NameBot gets near 0 recall - perhaps relatively more bots in the training data (or other data balancing techniques) would help it find more bots even in the fully imbalanced test data. Or conversely, even if "balanced everything including test set" isn't the most realistic and shouldn't be the "main" scenario, an experiment with that could provide evidence for/against whether the imbalance is playing a big role vs. performance drop generally due to bot evolution alone.
> >
> > It's understandable that the paper doesn't have a whole bunch of experiments like this and more generally trying different adjustments, hyperparameters, and so forth to optimize each model, since it's a very large number of models and probably infeasible computationally. But I do feel some more limited experiments could provide more evidence (and potentially insights) to explain the performance drop, which still remains very large (many models over 30 percentage points drop). For example, instead of all 35 models, perhaps taking a couple that collapse completely, a couple that get a lot worse but still above naive baseline, and a couple that still do relatively well. And experimenting with those in more depth to show the role played (or that there is no role played) by class imbalance, as well as any other hyperparameters or factors that seem potentially relevant.
> >
> > There's not much time though and even with fewer models that may take a while, so probably best left for future work. For the moment, I'd suggest briefly but explicitly drawing attention to the imbalance in the paper. Something along the lines that you wrote here, that the data was left imbalanced in the main experiment for increased realism. And that that may favor methods which can better handle it (which might explain some of the performance drop). Although it does raise more questions for future research, perhaps this is another benefit provided by this dataset? Unlike this one, according to e.g. Table 1, previous datasets were typically balanced or more bot than human, which is (at least in many contexts) unrealistic.
> >
> > Overall here, I appreciate that you plan to revisit the conclusions, though I'm not completely clear what revisions are planned (are those new 6.99% and 3.79% global average numbers from F1, and are you planning to base the main conclusions on that now? Or were those updated accuracy numbers?). From my perspective, if the imbalance and its potential role is more clear to the reader, and the conclusions are revised to either be based on F1 or to at least explain some of the similarities and/or differences between the accuracy and F1 results and what they may indicate, then that would mostly address my concerns here.

---

> > > ### Author Response · Authors · 2022-08-22
> > > **Thank you for your thoughtful insights regarding class imbalance.**
> > >
> > > Thank you for your detailed and thoughtful insights.
> > >
> > > - We agree that class imbalance would be an important topic in the future as bots and humans are not naturally balanced on social media (<5% according to Twitter [1], 9-15% according to an old survey [2]). It is still debatable what the best practices are while I personally would argue for a realist approach to use the data as-is in its own imbalanced way. Your comments successfully highlighted this issue to us and will continue to inspire our further research.
> > >
> > > - The 6.99% and 3.79% are calculated from the F1-scores. We are thinking about moving to using F1-score as the main evaluation metric while providing other metrics in the appendix. I hope to also hear from our project team when they return from their vacations before making this important change.
> > >
> > > [1] Musk’s Dispute With Twitter Over Bots Continues to Dog Deal. Kurt Wagner, Bloomberg. https://www.bloomberg.com/news/articles/2022-07-07/twitter-reiterates-that-spam-bots-are-well-under-5-of-users#:~:text=Twitter%20has%20repeatedly%20said%20that,confirmation%20about%20Twitter's%20bot%20percentage.
> > >
> > > [2] Yardi, Sarita, Daniel Romero, and Grant Schoenebeck. "Detecting spam in a twitter network." First monday (2010).

---

> ### Author Response · Authors · 2022-08-10
> **F1-score for Figure 3. (3/14)**
>
> ```
> "Similar concerns apply to other accuracy results, especially Figure 3, where no F1 results are reported, and the class imbalance for the sub-clusters may vary from the overall dataset and could be even more extreme."
> ```
>
> We saved the F1-score results of Figure 3 and would like to share them with the reviewer. However, the guy with the code for Figure 3 is away on vacation. I will personally hunt him down and present Figure 3 in F1-score once he returned, presumably in mid-August. Thanks for your understanding.

---

> > ### Comment · Reviewer_Mgrm · 2022-08-17
> > **Re: 3/14**
> >
> > OK, understandable. I'll definitely take a look if you're able to share these results. That said, in one of the new tables you shared on github you have information on the subcommunities that shows the classes there are balanced. Using the accuracy metric in that case is not so concerning, so my concerns for this particular experiment are generally already addressed by that, thank you.

---

> > > ### Author Response · Authors · 2022-08-22
> > > **Figure 3 with F1-score is available.**
> > >
> > > Figure 3 with F1-score is available now: https://github.com/LuoUndergradXJTU/TwiBot-22/blob/master/pics/figure3f1.jpg. Existing conclusions about Figure 3 still hold true, while there are minor differences in the specific numbers.

---

> ### Author Response · Authors · 2022-08-10
> **Potential bias coming from the annotation process. (4/14)**
>
> ```
> "Certain models are part of the label annotation process, i.e. lines 183-194. Could that potentially bias the benchmarking in favor of similar models? For example, RGCN is one of the models in the annotation process, and then BotRGCN is noted to perform particularly well in the benchmarking."
> ```
>
> This is an insightful observation. While BotRGCN does indeed perform well, random forest, though used as part of the labeling functions, does not perform very well on TwiBot-22. (Abreu et al. and Bot Hunter) In addition, while not used as part of the labeling function, Kantepe et al. do perform very well on TwiBot-22 in terms of F1-score. These instances suggest that the labeling process does not have systematic bias toward specific baseline methods.

---

> > ### Comment · Reviewer_Mgrm · 2022-08-17
> > **Re: 4/14**
> >
> > I think this reasoning makes sense. It's not ironclad though, so I think it may warrant further investigation in the future. One suggestion is to examine in some way how much each individual model is contributing to the labeling process. For example, it could be that although random forest is there, removing it would result in nearly the same labels, whereas including or removing RGCN changes the labels a lot. In that case, since RF has little impact, it probably would not produce much bias, while RGCN might produce more. On the other hand, if they have similar impact, then that would add more support for your argument here that there is no significant bias, since (among other examples you gave) RF performs poorly while RGCN performs well.
> > Another suggestion is using the manually labeled data as test set. I think I mentioned that in different context, but here if the performance and rankings of the benchmarked models are similar to the full TwiBot-22 results, that could provide fairly strong evidence the labels don't have any model-specific bias, since the human labels do not.

---

> > > ### Author Response · Authors · 2022-08-22
> > > **Experiment on the effect of individual labeling functions.**
> > >
> > > We remove each labeling function and re-do the Snorkel-based annotation process. We examine how many labels have changed and report the results on GitHub: https://github.com/LuoUndergradXJTU/TwiBot-22/tree/master/src/annotation. The results indicate that RGCN and random forest have a similar impact on the data annotation process (5.7% and 4.13% of labels changed as a result of removing RGCN and random forest).

---

> > > ### Author Response · Authors · 2022-08-22
> > > **Experiment on using manually labeled data as test set.**
> > >
> > > We use the training set and validation set in TwiBot-22 while using the following sets as the test set. We used 6 baseline methods for a quick evaluation.
> > >
> > > - **Test1**: using only the 1,000 manually annotated users in Section 3.2.
> > >
> > > - **Test2**: using only the 500 manually annotated users in Section 3.3.
> > >
> > > - **Test3**: using both the 1,000 and 500 manually annotated users.
> > >
> > > We report the results on GitHub: https://github.com/LuoUndergradXJTU/TwiBot-22#test1. There is a slight increase in overall model performance in comparison to using the full TwiBot-22 test set. GNN-based methods (GAT, BotRGCN, RGT) still generally outperform random forest-based methods (Moghaddam et al., SGBot, BotHunter) in terms of F1-score, which is compatible with the results when using the original TwiBot-22 test set.

---

> ### Author Response · Authors · 2022-08-10
> **Clarifying the value and meaning of $k$. (5/14)**
>
> ```
> "For the expansion with k users discussed on lines 151-160, I couldn't find any explicit value of k. The paper does discuss adding 6 users at a time in appendix A.2, so my best guess is k=2. But from lines 154-155, it seems that there should be k2 users added for a true-false metadata but k3 for others, i.e. different values added depending on whether true-false or not. I suggest in the appendix explicitly saying the value of k, and if indeed different numbers of users are added depending on true-false or not, saying the two total numbers added per step (e.g. "we add 6 users per step, except for true-false metadata where we add 4"). If it's always 6 users regardless of true-false or not, then I suggest making k the total number of users added, and rewriting 151-160 in terms of fraction of k (e.g. k/3 for each of highest/lowest/random on line 154, k/2 for true-false on line 155, etc.)."
> ```
>
> k=6. We will clearly state this in the appendix. It is always 6 users for both strategies, we will clearly state that k/3 for each of highest/lowest/random on line 154, k/2 for true-false on line 155 in the final version.

---

> ### Author Response · Authors · 2022-08-10
> **Tweet collection based on hashtags. (6/14)**
>
> ```
> "When collecting additional tweets related to a hashtag, e.g. lines 597-598, does this include all tweets that include that hashtag during the collection period? Or only some subset, e.g. a fixed amount per hashtag?"
> ```
>
> We used the search feature of the Twitter API to collect *at most* 100 tweets related to each hashtag.

---

> ### Author Response · Authors · 2022-08-10
> **More on the 10 sub-communities. (7/14)**
>
> ```
> "In the generalization study, it would be helpful to elaborate further (e.g. in appendix) on exactly how the subcommunities are constructed. For the user-based ones, are they the one-hop neighborhoods (based on follow relation?) of the 5 selected users? For the hashtag clustering ones, how is word2vec applied, and is anything special done to make it work with hashtags (which are often not dictionary words)? When applying K-means to those representations, is K=5, or else what is the value and how was it chosen? Is a user included in the sub-community as long as they use a hashtag within it, or is there a stricter criterion?"
> ```
>
> For the user-based ones, they are multi-hop neighborhoods based on follow relations of the starting user.
>
> For the hashtag ones, we remove the special character `#` and use pre-trained word2vec to encode hashtag text. Hashtags that are not dictionary words are not included in the process.
>
> Yes, K=5. Yes, a user is included in the sub-community if the user tweets about a hashtag within it. In the case of too many users in one sub-community, we conduct random selection to ensure the sub-communities are of desirable size.

---

> > ### Comment · Reviewer_Mgrm · 2022-08-17
> > **Re: 7/14**
> >
> > For user-based, how do you decide how many hops? I see from one of your new tables on github that it's 5000 humans and 5000 bots in each sub-community, which is helpful information, thank you for making that available. But I'm still not completely clear how they're chosen. Within X hops and then random selection to get the right totals?
> >
> > Incidentally, super minor, but the link to the paper on the sub-community table page https://github.com/LuoUndergradXJTU/TwiBot-22/blob/master/descriptions/sub-communities.md in fact links to the descriptions folder rather than the paper.
> >
> > For the hashtags, that is clear now, thanks for explaining. I do wonder how much only dictionary words (and specifically a dictionary not designed for twitter hashtags) might influence the clustering. It seems common to have hashtags that are e.g. multiple words strung together. That said, since the experiment is about generalization of bot detection methods, and I don't see any obvious reason non-dictionary hashtags would correlate with that, it probably has minimal impact on the results here. So perhaps something for future work but your approach here does seem reasonable for the current scope.

---

> > > ### Author Response · Authors · 2022-08-22
> > > **Following-up on the 10 sub-communities.**
> > >
> > > For each user expansion, we select 20% of all the user's neighbors and add it to the set. We repeat the BFS-like process until both humans and bots reach 5,000. We then conduct random selection to fix the numbers to 5,000 humans and 5,000 bots. We will add this explanation to Appendix B.5 to facilitate better understanding.
> > >
> > > Thank you for pointing out this minor mistake. Also, thank you for agreeing that our current approach to hashtags is reasonable.

---

> ### Author Response · Authors · 2022-08-10
> **Performance difference between line 199 and Figure 2(c). (8/14)**
>
> ```
> "On line 199 the labels achieve 90.5% accuracy compared to the expert hand-labeled test set. But then in figure 2c the paper reports accuracy against a different expert hand-labeled set and it's a bit under 80%. This difference seems somewhat large to be random? Do the authors have any comments, explanation, or hypothesis here? Possibly related, it is not clear to me how the 5-way labels the experts provide (in the second case) are converted to the 2-way labels in the dataset to make this comparison. How are disagreements resolved between the 3 experts providing a label for each example? Similar question for the 1000 expert annotations used in the dataset construction itself. The paper explains each example is labeled by 5 experts as human/bot/not sure, and then majority voting is applied, but how are cases without a majority (e.g. 2x human 2x not sure 1x bot) resolved? What happens to cases with majority not sure, are they dropped, and if so is the 1000 the remainder after dropping such cases or is the actual number used somewhat less than 1000?"
> ```
>
> There is indeed a performance discrepancy between line 199 and Figure 2c. We believe the difference could be attributed to:
>
> - Different purposes: In line 199, those expert labels were adopted to tune the hyperparameters of the Snorkel-based data annotation system. In Figure 2c, those expert labels were adopted to evaluate the label quality of TwiBot-20 and TwiBot-22. Since the former set is tuned on, it is reasonable that the results are higher. Similar to the suggestion of reviewer zYJs, we will change the name of the latter to "annotation quality evaluation test" to better differentiate them.
>
> - Different experts: 17 experts participated in the expert labels in line 199, where the criteria are “familiar with bot detection literature, and have conducted experiments with the TwiBot-20 datasets.” (Line 604-605) In contrast, only 6 researchers participated in the expert labels in Figure 2c, where the criteria are “familiar with Twitter bot detection research and most of them have previously published on this topic.” (Line 613-614) The bar of the latter is considerably higher, thus we believe the lower performance in Figure 2c could be the result of an improved expert cohort.

---

> > ### Comment · Reviewer_Mgrm · 2022-08-17
> > **Re: 8/14**
> >
> > I see, those reasons do seem plausible explanations why performance would be better on the one set vs. the other.
> >
> > I'm still a bit unsure how disagreements were resolved, e.g. cases without a majority. It would be helpful to address that in the paper (probably appendix). This could, for example, help inform researchers doing similar labeling in the future.

---

> > > ### Author Response · Authors · 2022-08-22
> > > **Resolving disagreement in expert annotations.**
> > >
> > > For the first expert label set with 17 annotators, there would always be a majority. For the second expert label set with 6 experts, we ask 1 more annotator to break the tie and inform the team of the decision, while a discussion is held at the end for cases where someone objects to the majority ruling to reach a consensus. Out of the 500 annotated users in the second set, only 17 needed a tie-breaking vote.
> > >
> > > In addition, we provide the individual annotation data on GitHub (https://github.com/LuoUndergradXJTU/TwiBot-22/blob/master/src/annotation/record_combine.csv) since this information could be interesting and further explored.

---

> ### Author Response · Authors · 2022-08-10
> **Clarifying F1-score calculation. (9/14)**
>
> ```
> "In Figure 2c, is the F1 reported for the positive class "bot" or something else e.g. macro-average? I would guess that the results in Table 8 are F1 for the positive class "bot" but would also be good to confirm."
> ```
>
> Yes, the F1-score is calculated for the positive class bot.

---

> ### Author Response · Authors · 2022-08-10
> **Fixing typos. (10/14)**
>
> ```
> "There are a few typos such as:
> Line 163: "stops until the user network contains..." -> should read "stops when the ..." or "continues until the ..."
> Line 200: imporved -> improved
> Line 221: bot reporistory -> bot repository
> Line 599: period missing at end of sentence"
> ```
>
> Thank you for your detailed reading. We will correct them in the final version.

---

> ### Author Response · Authors · 2022-08-10
> **Application for accessing TwiBot-22. (11/14)**
>
> ```
> "For example, a number of datasets require a brief application form for access; could something like that help ensure this data is used for academic research purposes, or would that limit accessibility more than it is worth?"
> ```
>
> For our previous work TwiBot-20, we introduced an application process and we plan to use it for TwiBot-22 as well. Firstly, one needs to contact the TwiBot-22 team via an email address that belongs to an academic or research institution. We will then ask the individual to provide relevant information such as project description, research advisor, and intended use. After that, a panel from the TwiBot-22 team would convene once a week to discuss these applications and approve them only if the applicant successfully demonstrates that their purpose is research-only. We hope that in this way bot operators and malicious actors would be prohibited from TwiBot-22 and mitigate its societal impact.

---

> ### Author Response · Authors · 2022-08-10
> **Make expert annotations publicly available. (12/14)**
>
> ```
> "Two expert hand-labeled subsets of users were collected during the annotation process. It might be interesting to see results of the different benchmarked models on these subsets, since in theory they are the most accurate labels available. Also, are they available or marked somewhere in the dataset (on github/drive/etc.)? I didn't find them, but only looked briefly."
> ```
>
> Thank you for your suggestion. We have now provided these two expert label sets in the GitHub repository:
>
> https://github.com/LuoUndergradXJTU/TwiBot-22/tree/master/src/annotation

---

> ### Author Response · Authors · 2022-08-10
> **The potential of Snorkel and mixture of experts on bot detection. (13/14)**
>
> ```
> "The Snorkel model used for annotation seems like it might have better performance than any benchmarked model, at least according to the first set expert labels where it achieved 90.5% accuracy. Not crucial to the paper, but besides annotation, is it worth considering that as a future method? Would it perform well on the other datasets benchmarked here?"
> ```
>
> This is a very interesting direction and we are currently thinking of working on it. Similar to Snorkel, we are working to leverage mixture of experts to fuse different bot detection models and make the predictions more stable. I hope this future work would generate interesting findings, but at the moment that’s all I have to share.

---

> ### Author Response · Authors · 2022-08-10
> **Thank you very much. (14/14)**
>
> ```
> "Regardless of the final decision here, I appreciate the author's contributions to this important area and hope they will continue research along these lines. Thank you!"
> ```
>
> Thank you for your constructive feedback. This is one of the most extensive reviews I have ever received. It was a pleasure mulling over every one of your suggestions and I’m sure that the paper will be much stronger thanks to your time and efforts! :)

---

### Official Review · Reviewer_SrxK · 2022-07-27
**A well-documented framework with benchmarks.**

**Rating:** 6
**Confidence:** 3
**Clarity:** The paper is clearly written.

**Strengths:**

(1) This paper significantly and robustly extends the data available in Twitter bot detection. It proposes a dataset 5 times larger than the previous dataset.

(2) The dataset contains rich information for Twitter bot detection, providing for the first time ever, a heterogeneous graph.

(3) Extensive experiments are conducted, and the results provide insight into future work. Also, a well-organized evaluation framework is proposed to allow easy access and reuse of datasets and all baselines.

**Weaknesses:**

(1) The user expansion process described in Section A.2 is not clear. The authors clarified that the number of users increased from the starting user by expanding to their following and follower users. However, how to achieve 1 million users consequently from starting users? For example, how many iterations of random selection are performed? Did the target number of users exist? More details are needed for clarity.

(2) In Section 3.3, annotation quality comparisons are conducted between Twibot-22 and Twibot-20 only. However, a comparison with other baseline datasets such as cresci-15 should be considered for fair comparison. A comparison of the quality with all other baseline datasets should also be presented that verify whether the proposed dataset has high-quality annotation labels compared to others.

(3) In Section 4.4, the generalization study, the criteria for selecting the five users (e.g., @ BarackObama, @elonmusk, …) for constructing sub-communities should be clarified. For example, what exactly is the “interest domain” used as the criteria? How are the users of the sub-community distributed? Do the characteristics of the sub-communities significantly statistically differ? If so, how the authors evaluate the differences? The authors should provide more details and discussion for verifying that 10 sub-communities could represent the real-world data.



**Additional Feedback:**

Line 532: add a period.

**Correctness:**

The database seems quite soundly constructed, and evaluation methods and experiment design are appropriate and performed correctly.

**Documentation:**

This paper provides instructions on the URL where data can be found with detailed descriptions and applications.

**Ethics:**

No. The author discussed the issues in Section A.6

**Relation To Prior Work:**

The paper details what previous datasets or approaches for detecting Twitter bots exist and how these datasets differ from the proposed dataset’s aspects of scale, graph type, and the quality of annotations.

**Summary And Contributions:**

This paper proposes a large graph-based dataset for Twitter bot detection that helps prevent misinformation and maintain the integrity of the online discourse. Twibot-22, proposed by the authors, includes five times more users than the existing dataset and is a heterogeneous information network, including 4 entities and 14 relation types. It also uses a weak supervision learning strategy to generate high-quality annotations at a low cost. The authors conduct 35 benchmark tests on the proposed dataset. Experimental results imply that Twibot-22 can be used to develop improved automatic methods optimized for Twitter bot detection in more complex real-world situations.

---

> ### Author Response · Authors · 2022-08-10
> **More on the user expansion process. (1/3)**
>
> Thank you for your detailed and thoughtful review of our submission: *TwiBot-22: Towards Graph-Based Twitter Bot Detection*. We hereby address the raised questions and concerns. We believe the paper would be much stronger thanks to your feedback.
>
> ```
> “The user expansion process described in Section A.2 is not clear. The authors clarified that the number of users increased from the starting user by expanding to their following and follower users. However, how to achieve 1 million users consequently from starting users? For example, how many iterations of random selection are performed? Did the target number of users exist? More details are needed for clarity.”
> ```
>
> In the beginning, TwiBot-22 only contains one user (the starting user). We then use the Twitter API to randomly retrieve 1,000 followers and 1,000 followees of the user. To control the homophilic behavior on social networks, we use diversity-aware sampling to select 6 users from the 2,000 users (1,000 followers + 1,000 followees), making sure that these 6 users are very different from the starting user. At the end of one iteration, there are 7 users in TwiBot-22.
>
> At the beginning of the second iteration, we choose 1 user from the unexpanded users in TwiBot-22 (where there are 6) to go over the above process. In the end, a total of 10,509 iterations are performed. We will add this clarification to the paper.

---

> ### Author Response · Authors · 2022-08-10
> **Assessing annotation quality for other datasets. (2/3)**
>
> ```
> "In Section 3.3, annotation quality comparisons are conducted between Twibot-22 and Twibot-20 only. However, a comparison with other baseline datasets such as cresci-15 should be considered for fair comparison. A comparison of the quality with all other baseline datasets should also be presented that verify whether the proposed dataset has high-quality annotation labels compared to others."
> ```
>
> Thank you for your suggestion. Since some of the bot detection experts we relied on are currently on vacation, we could not provide these results in time. We will set the wheels in motion once they came back from vacation.

---

> ### Author Response · Authors · 2022-08-10
> **More on the 10 sub-communities. (3/3)**
>
> ```
> "In Section 4.4, the generalization study, the criteria for selecting the five users (e.g., @ BarackObama, @elonmusk, …) for constructing sub-communities should be clarified. For example, what exactly is the “interest domain” used as the criteria? How are the users of the sub-community distributed? Do the characteristics of the sub-communities significantly statistically differ? If so, how the authors evaluate the differences? The authors should provide more details and discussion for verifying that 10 sub-communities could represent the real-world data."
> ```
>
> The idea of the “interest domain” comes from TwiBot-20, where researchers collected users from politics, business, entertainment, and sports to provide a diversified set of users. In the generalization study, we largely follow this principle to select *@BarackObama* (politics), *@elonmusk* (business), *@cnn* (news), *@NeurIPSConf* (academics), and *@ladygaga* (entertainment). We agree that this selection is a bit casual and should serve only as a preliminary evaluation of bot detection generalization. We leave the theoretical analysis of the generalization ability of bot detection algorithms to future work.

---

> ### Author Response · Authors · 2022-08-22
> **Follow-up on author response.**
>
> Thank you for your valuable assessment and constructive feedback. We addressed the raised concerns and we believe the paper is much stronger thanks to your feedback. We wonder if it might be possible to go through our response and see if there's anything else we could do to improve this work. Thank you in advance.

---

### Official Review · Reviewer_9eKz · 2022-07-27
**A well evaluated graph based approach to identify twitter bots**

**Rating:** 7
**Confidence:** 4
**Clarity:** Paper is well written. Please address…

**Strengths:**

The following are the strengths of the paper

1) Using labeling functions to create labels and utilizing weak supervision approach. This demonstrates scalability
2) The dataset is useful as there is an immense need for bot detection techniques with increase in social media research
3) Extensive evaluation with several existing datasets to draw comparison with the graph based method

**Weaknesses:**

The following are the weakness. These needs to be addressed to strengthen the paper

1) TwiBot-22 has less performance when compared to TwiBot-20. Authors made the statement on the decrease in the performance but have not discussed potential reasons for decrease in the performance. Is it because of adding more noise through labelling functions?
2) The authors do not discuss any limitations of the approach or the dataset. A few limitations could add more perspective to the paper
3) More details on the dataset is required. For example in Table 1, the authors list number of users and number of humans and number of bots in the dataset. But the authors fail to mention how many of tweets are actually bot tweets. This would be a great contribution factor and add value to the dataset if there are more bot tweets than user tweets. Tweet distribution must be added to the paper.

**Additional Feedback:**

Authors need to add more information on the dataset and the distribution of tweets in the dataset.

**Correctness:**

The claims made in the paper are correct and the dataset is constructed in a sound way. The benchmark evaluation methods and experiment design are appropriate and performed correctly.

**Documentation:**

Yes

**Ethics:**

I dont see any ethical concerns

**Relation To Prior Work:**

Yes. Prior work is well discussed

**Summary And Contributions:**

This paper utilizes a graph based approach to create the largest Twitter bot detection dataset. The primary strength of the paper is the dataset which contains over 100,000 bot users which is very useful for research on social media data. Additionally, the authors perform extensive evaluation using latest methods and re-implemented previous baselines and compared the performance of their dataset. The following are the contributions.

1) A graph based approach, TwiBot-22, which contains bot detection dataset that establishes the largest benchmark
2) benchmark framework and re-implemented 35 baselines for evaluation
3) code available to reproduce and dataset available for research

---

> ### Author Response · Authors · 2022-08-10
> **Performance comparison between TwiBot-20 and TwiBot-22. (1/3)**
>
> Thank you for your detailed and thoughtful review of our submission: *TwiBot-22: Towards Graph-Based Twitter Bot Detection*. We hereby address the raised questions and concerns. We believe the paper would be much stronger thanks to your feedback.
>
> ```
> "TwiBot-22 has less performance when compared to TwiBot-20. Authors made the statement on the decrease in the performance but have not discussed potential reasons for decrease in the performance. Is it because of adding more noise through labelling functions?"
> ```
>
> We believe that the performance drop from TwiBot-20 to TwiBot-22 could be attributed to “bot evolution” [1,2,3,4], where Twitter bots are constantly evolving to evade detection, making the performance of existing bot detectors worse over time. Since TwiBot-22 is the most recent dataset, we believe it is reasonable that model performance on TwiBot-22 is lower than TwiBot-20, which was proposed two years ago. In addition, this performance discrepancy suggests that “Twitter bot detection is still an open problem that calls for further research” (Line 257), further proving the value and importance of TwiBot-22. We will add this discussion to the final version.
>
> [1] Yang, Chao, Robert Harkreader, and Guofei Gu. "Empirical evaluation and new design for fighting evolving twitter spammers." IEEE Transactions on Information Forensics and Security 8.8 (2013): 1280-1293.
>
> [2] Cresci, Stefano. "A decade of social bot detection." Communications of the ACM 63.10 (2020): 72-83.
>
> [3] Cresci, Stefano, et al. "The paradigm-shift of social spambots: Evidence, theories, and tools for the arms race." Proceedings of the 26th international conference on world wide web companion. 2017.
>
> [4] Feng, Shangbin, et al. "BotRGCN: Twitter bot detection with relational graph convolutional networks." Proceedings of the 2021 IEEE/ACM International Conference on Advances in Social Networks Analysis and Mining. 2021.

---

> ### Author Response · Authors · 2022-08-10
> **Expand on limitations discussion. (2/3)**
>
> ```
> "The authors do not discuss any limitations of the approach or the dataset. A few limitations could add more perspective to the paper."
> ```
>
> We discussed one limitation in lines 642-645. We hereby expand our discussion of TwiBot-22 limitations and we will add it to the final version.
>
> - One minor limitation of TwiBot-22 is that we do not download and store user media (images and videos) in TwiBot-22, while these multimedia content might be useful for bot detection. However, if researchers do deem multimedia content as necessary for bot detection, they can download with media links in TwiBot-22 by themselves.
>
> - Another limitation of TwiBot-22 is the data annotation strategy. While we made our best efforts to select representative labeling functions and adopt Snorkel for weak supervision-based annotation, the labels are not golden and may be subject to minor noise.
>
> - In addition, TwiBot-22 only collected 40 tweets from a single user, while certain time-based methods require the whole tweeting history of a user for bot detection. We chose to collect 40 tweets based on [1], which suggests that 40 tweets would be a good approximation and facilitate stable bot detection algorithms.
>
> [1] Ng, Lynnette Hui Xian, Dawn C. Robertson, and Kathleen M. Carley. "Stabilizing a supervised bot detection algorithm: How much data is needed for consistent predictions?." Online Social Networks and Media 28 (2022): 100198.

---

> ### Author Response · Authors · 2022-08-10
> **More dataset details. (3/3)**
>
> ```
> "More details on the dataset is required. For example in Table 1, the authors list number of users and number of humans and number of bots in the dataset. But the authors fail to mention how many of tweets are actually bot tweets. This would be a great contribution factor and add value to the dataset if there are more bot tweets than user tweets. Tweet distribution must be added to the paper."
> ```
>
> There are 6,967,355 bot tweets and 81,250,102 tweets from genuine users. We will add this information to the final version. Let me know if you believe any other information would be helpful. :)

---

> ### Author Response · Authors · 2022-08-22
> **Follow-up on author response.**
>
> Thank you for your valuable assessment and constructive feedback. We addressed the raised concerns and we believe the paper is much stronger thanks to your feedback. We wonder if it might be possible to go through our response and see if there's anything else we could do to improve this work. Thank you in advance.

---

### Official Review · Reviewer_zYJs · 2022-07-28
**Recommending accept after the discussion phase**

**Rating:** 7
**Confidence:** 4
**Clarity:** Yes. The flow of the paper is good.

**Strengths:**

A. The dataset improves upon the existing graph-based datasets in terms of dataset scale, graph structure and annotation quality.
B. The evaluation framework which includes implementation of many baseline algorithms and few existing datasets can facilitate bot detection related research.


**Weaknesses:**

A. Regarding the first stage of data collection process, this dataset appears to be a direct extension of a previous dataset “TwitBot20”[1] by the same authors with a minor change in the "diversity-aware" sampling.

B. Line 198 : "...We further evaluate annotation quality on the test set of expert annotations...". The test set implied here seems to correspond to the 200 users' set mentioned in line 181. However, in appendix A.5, the test set seems to consist of 1000 users (500 from TwitBot20 and 500 from TwitBot22). Need to address this ambiguity.

C. In section 4.4 (Generalization Study), more details about the 10 sub-communities could be included (similar to what is done in Table 1 for the whole dataset). This could be useful in inspecting the results presented in Figure 3.

D. The authors have stated a possible limitation to be the lack of multimedia content in the dataset. But they have indicated that it is possible to do so using media links in TwitBot-22. The solution being alluded to is not clear.

[1] Shangbin Feng, Herun Wan, Ningnan Wang, Jundong Li, and Minnan Luo. TwitBot-20: A comprehensive twitter bot detection benchmark. In Proceedings of the 30th ACM International Conference on Information & Knowledge Management, pages 4485-4494, 2021.


**Additional Feedback:**

N/A

**Correctness:**

Yes. Most of the claims are supported by evidence. However, it is not clear to me that how exactly future graph-based algorithms can leverage a more complex entity-relation structure of the dataset.

**Documentation:**

The provided URL has both the current dataset and the prior dataset.

**Ethics:**

No aparent concern.

**Relation To Prior Work:**

This is not the first graph-based dataset. They have mentioned prior graph-based datasets and provided a comparison.

**Summary And Contributions:**

This paper proposes a new graph-based Twitter bot detection benchmark dataset. This dataset has been presented as an improvement over the already existing graph-based datasets in terms of dataset scale, extensiveness of the graph structure and annotation quality. The authors evaluated the proposed dataset, along with 8 more existing datasets, using 35 twitter bot detection baseline algorithms. Based on evaluation, the authors argue that the graph-based component of the sota algorithms is responsible for their better performance hence the need of an extensive graph-based dataset. Finally, the authors provide a publicly available evaluation framework which includes all the datasets and baselines used in the paper.

---

> ### Author Response · Authors · 2022-08-10
> **What's new in TwiBot-22 data collection? (1/5)**
>
> Thank you for your detailed and thoughtful review of our submission: *TwiBot-22: Towards Graph-Based Twitter Bot Detection*. We hereby address the raised questions and concerns. We believe the paper would be much stronger thanks to your feedback.
>
> ```
> "Regarding the first stage of data collection process, this dataset appears to be a direct extension of a previous dataset “TwitBot20”[1] by the same authors with a minor change in the "diversity-aware" sampling."
> ```
>
> Indeed, TwiBot-22 employs a similar BFS-based approach to TwiBot-20. One key distinction between the data collection process in TwiBot-22 and Twibot-20 is the diversity-aware sampling, which aims to address the homophilic behavior prevalent in graph datasets. (which reviewer vUh8 found as “robust” and “important”) We will highlight the relation between our diversity-aware sampling strategies and previous literature in graph homophily [1,2,3] in the final version to better make our case.
>
> [1] Guerrero-Solé, Frederic. "Interactive behavior in political discussions on Twitter: Politicians, media, and citizens’ patterns of interaction in the 2015 and 2016 electoral campaigns in Spain." Social Media+ Society 4.4 (2018): 2056305118808776.
>
> [2] Solomon, R. Sudhesh, et al. "Understanding the psycho-sociological facets of homophily in social network communities." IEEE Computational Intelligence Magazine 14.2 (2019): 28-40.
>
> [3] Colleoni, Elanor, Alessandro Rozza, and Adam Arvidsson. "Echo chamber or public sphere? Predicting political orientation and measuring political homophily in Twitter using big data." Journal of communication 64.2 (2014): 317-332.

---

> > ### Comment · Reviewer_zYJs · 2022-08-22
> > **Satisfied with the author response**
> >
> > Thanks a lot for your detailed response to all the reviews. Going over the subsequent discussions, I believe that the revised version has seen a lot of improvement compared to the original. I am revising my score to accept.

---

> ### Author Response · Authors · 2022-08-10
> **Clarifying the difference between the two expert annotation sets. (2/5)**
>
> ```
> "Line 198 : "...We further evaluate annotation quality on the test set of expert annotations...". The test set implied here seems to correspond to the 200 users' set mentioned in line 181. However, in appendix A.5, the test set seems to consist of 1000 users (500 from TwitBot20 and 500 from TwitBot22). Need to address this ambiguity."
> ```
>
> We would like to clarify that line 198 and appendix A.5 actually involve two different “test sets”.
>
> In section 3.2 (which includes lines 198 and 181), the 1000-user set is selected from TwiBot-22 alone, annotated by experts, and an 8:2 split is created to train labeling functions and enable weak supervision-based data annotation.
>
> In appendix A.5, the “test set” refers to the one used in the annotation quality study on lines 211-218. This 1000-user test set is selected from both TwiBot-20 and TwiBot-22, evaluated by researchers, and used to compare the annotation quality between TwiBot-22 and TwiBot-20.
>
> We understand that this ambiguity might be the result of excessively using the term “test set”. We will change the one in appendix A.5 and lines 211-218 to "annotation quality evaluation set" to avoid future confusion.

---

> ### Author Response · Authors · 2022-08-10
> **Statistics of the 10 sub-communities. (3/5)**
>
> ```
> "In section 4.4 (Generalization Study), more details about the 10 sub-communities could be included (similar to what is done in Table 1 for the whole dataset). This could be useful in inspecting the results presented in Figure 3."
> ```
>
> We hereby present the statistics of the 10 sub-communities. We will add it to the appendix in the final version.
>
> https://github.com/LuoUndergradXJTU/TwiBot-22/blob/master/descriptions/sub-communities.md

---

> ### Author Response · Authors · 2022-08-10
> **TwiBot-22 and Twitter media content. (4/5)**
>
> ```
> "The authors have stated a possible limitation to be the lack of multimedia content in the dataset. But they have indicated that it is possible to do so using media links in TwitBot-22. The solution being alluded to is not clear."
> ```
>
> TwiBot-22 contains Twitter media links thanks to the new Twitter API v2. Since images and videos [1,2] are important sources of online misinformation, we suggest that future bot detection research could leverage multimedia content for robust bot detection. While we did not download media content and store it in TwiBot-22, researchers could use the media links provided in TwiBot-22 to retrieve them by themselves. We will add this clarification to the final version.
>
> [1] Huh, Minyoung, et al. "Fighting fake news: Image splice detection via learned self-consistency." Proceedings of the European conference on computer vision (ECCV). 2018.
>
> [2] Lemos, André Luiz Martins, Elias Cunha Bitencourt, and João Guilherme Bastos dos Santos. "Fake news as fake politics: the digital materialities of YouTube misinformation videos about Brazilian oil spill catastrophe." Media, Culture & Society 43.5 (2021): 886-905.

---

> ### Author Response · Authors · 2022-08-10
> **Why is complex network structure necessary? (5/5)**
>
> ```
> "However, it is not clear to me that how exactly future graph-based algorithms can leverage a more complex entity-relation structure of the dataset."
> ```
>
> The motivation to provide and leverage a more complex network structure is grounded in the research and developments of heterogeneous graph neural networks [1,2,3]. Different from homogeneous GNNs, heterogeneous GNNs could leverage graph data with multiple types of nodes and edges, resulting in better performance and improved modeling of real-world networks. In fact, heterogeneous GNNs have also shown promising performance in bot detection research but are also limited by the incomplete graph structure [4]. As a result, we aim to provide more types of entities and relations to construct a more heterogeneous graph, which has broad applications for both heterogeneous GNNs and bot detection research.
>
> [1] Hu, Ziniu, et al. "Heterogeneous graph transformer." Proceedings of The Web Conference 2020. 2020.
>
> [2] Lv, Qingsong, et al. "Are we really making much progress? Revisiting, benchmarking and refining heterogeneous graph neural networks." Proceedings of the 27th ACM SIGKDD Conference on Knowledge Discovery & Data Mining. 2021.
>
> [3] Schlichtkrull, Michael, et al. "Modeling relational data with graph convolutional networks." European semantic web conference. Springer, Cham, 2018.
>
> [4] Feng, Shangbin, et al. "Heterogeneity-aware twitter bot detection with relational graph transformers." Proceedings of the AAAI Conference on Artificial Intelligence. Vol. 36. No. 4. 2022.

---

> ### Author Response · Authors · 2022-08-22
> **Follow-up on author response.**
>
> Thank you for your valuable assessment and constructive feedback. We addressed the raised concerns and we believe the paper is much stronger thanks to your feedback. We wonder if it might be possible to go through our response and see if there's anything else we could do to improve this work. Thank you in advance.

---

### Official Review · Reviewer_vUh8 · 2022-07-28
**Good benchmark for Twitter bot detection**

**Rating:** 7
**Confidence:** 4
**Clarity:** The paper is very well-written and ea…

**Strengths:**

The paper has the following strengths:
1. It tackles the very relevant problem of bot detection on Twitter, which is of consequence to various downstream objectives like spread of misinformation, propagation of rumors, etc.
2. The dataset released is heterogenous with 4 different node types and 14 different relation types, making the dataset close to real-world graphs.
3. Extensive experiments are conducted with a range of state of the art methods, including graph neural networks, to demonstrate the usefulness of the released dataset. The experiments also cover interesting ablation studies to demonstrate the usefulness of having graph signals.

**Weaknesses:**

1. In the experiments section, I would have liked to see testing for statistical significance as well.

**Additional Feedback:**

None

**Correctness:**

I found the data sampling methodology to be robust, specially accounting for distribution and value diversity. Here, the latter relates to the concept of homophilic behavior, which is important to control for when creating graph datasets.

**Documentation:**

The documentation is sufficient.

**Ethics:**

My main concern here is whether the users in the dataset can be de-anonymized? And if yes, how do we handle the deletion of their profiles or tweets.

**Relation To Prior Work:**

I found the coverage of prior work to be sufficient. In fact, there are very good comparisons made to the existing TwiBot-20 benchmark in terms of data and modeling trends.

**Summary And Contributions:**

The paper tackles the problem of bot detection on Twitter with graph-based methods. To that effect, the paper makes the following contributions:
1. Creates an annotated graph dataset, TwiBot-22, of human and bot Twitter users, with 1M users in total.
2. Presents an evaluation framework to standardize the measurement interface across various different datasets and methods.

---

> ### Author Response · Authors · 2022-08-10
> **Statistical significance testing. (1/3)**
>
> Thank you for your detailed and thoughtful review of our submission: *TwiBot-22: Towards Graph-Based Twitter Bot Detection*. We hereby address the raised questions and concerns. We believe the paper would be much stronger thanks to your feedback.
>
> ```
> "In the experiments section, I would have liked to see testing for statistical significance as well."
> ```
>
> Thank you for your suggestion. We performed a test for statistical significance on each of the baselines and obtained the following results ($p=0.1$). The intervals in the following sheets are confidence intervals with statistical significance level $\alpha=0.1$, assuming the results conform to a t-distribution. Specially, we use `scipy.stats.t` to calculate the confidence interval under certain significance level and ignore those deterministic baselines (e.g. Botometer). The results for each metric are listed below.
>
> - accuracy: https://github.com/LuoUndergradXJTU/TwiBot-22/blob/master/descriptions/tss/acc.md
>
> - F1-score: https://github.com/LuoUndergradXJTU/TwiBot-22/blob/master/descriptions/tss/f1.md
>
> - precision: https://github.com/LuoUndergradXJTU/TwiBot-22/blob/master/descriptions/tss/precision.md
>
> - recall: https://github.com/LuoUndergradXJTU/TwiBot-22/blob/master/descriptions/tss/recall.md

---

> > ### Comment · Reviewer_vUh8 · 2022-08-10
> > **Thank you for the response.**
> >
> > Thank you for the additional analysis, this is convincing, please add it to the paper.

---

> > > ### Author Response · Authors · 2022-08-10
> > > **Will do.**
> > >
> > > I am thinking about creating a new section in the appendix (section C: Additional Experiment Results) to present significance testing results, precision, recall, and other results that the reviewers proposed to add. Let me know if that would be a good idea.
> > >
> > > Also, if you feel like your concerns are adequately addressed, I wonder if it is possible for you to update your score to reflect that. That would be very helpful. :)

---

> ### Author Response · Authors · 2022-08-10
> **More on data sampling methodology. (2/3)**
>
> ```
> "I found the data sampling methodology to be robust, specially accounting for distribution and value diversity. Here, the latter relates to the concept of homophilic behavior, which is important to control for when creating graph datasets."
> ```
>
> We agree that the two diversity-aware sampling techniques adopted in TwiBot-22 are very important. We will further ground them in previous literature about homophilic behavior and graph homophily [1,2,3].
>
> [1] Guerrero-Solé, Frederic. "Interactive behavior in political discussions on Twitter: Politicians, media, and citizens’ patterns of interaction in the 2015 and 2016 electoral campaigns in Spain." Social Media+ Society 4.4 (2018): 2056305118808776.
>
> [2] Solomon, R. Sudhesh, et al. "Understanding the psycho-sociological facets of homophily in social network communities." IEEE Computational Intelligence Magazine 14.2 (2019): 28-40.
>
> [3] Colleoni, Elanor, Alessandro Rozza, and Adam Arvidsson. "Echo chamber or public sphere? Predicting political orientation and measuring political homophily in Twitter using big data." Journal of communication 64.2 (2014): 317-332.

---

> > ### Comment · Reviewer_vUh8 · 2022-08-10
> > **Thank you for the response.**
> >
> > Yes, please do cite the works to ground your reasoning in existing research. There is another paper that talks about the same aspects of graph based modeling, please do cite that too: https://aclanthology.org/2021.findings-emnlp.287/

---

> > > ### Author Response · Authors · 2022-08-10
> > > **Thank you for your suggestion.**
> > >
> > > This paper is closely related to graph-based social network analysis and we will discuss it in the related work. Thank you for your suggestion!

---

> ### Author Response · Authors · 2022-08-10
> **Anonymize TwiBot-22. (3/3)**
>
> ```
> "My main concern here is whether the users in the dataset can be de-anonymized? And if yes, how do we handle the deletion of their profiles or tweets."
> ```
>
> Yes, the users in TwiBot-22 can and will be anonymized. We will remove the usernames and screen names from the TwiBot-22 in the final release upon acceptance. We agree that the deletion of user tweets is an important issue, and we will follow previous works such as [1] to provide a workaround: we will encode user tweets and descriptions with different pre-trained language models and release the extracted features. In this way, user privacy is protected while researchers could still leverage tweet content for robust bot detection.
>
> [1] Dritsa, Konstantina, et al. "A Greek Parliament Proceedings Dataset for Computational Linguistics and Political Analysis."

---

> > ### Comment · Reviewer_vUh8 · 2022-08-10
> > **Thank you for the response.**
> >
> > Another good way to handle deletion would be to release the dataset via an API that periodically checks if the tweet ID for a particular tweet is alive or not. If not, then the tweet can't be accessed.

---

> > > ### Author Response · Authors · 2022-08-10
> > > **Thank you for your suggestion.**
> > >
> > > This is an interesting approach and I have never considered something like this before. I am thinking that maybe we should filter the tweets and provide a new version once every month. One potential concern here is that bots tend to delete their malicious tweets after a period of time to avoid detection, which might cause problems for TwiBot-22. I will further discuss this with the team. Anyway, thank you for actively engaging with us!

---

### Official Review · Reviewer_4AAD · 2022-07-30
**Twitter dataset bot detection review**

**Rating:** 7
**Confidence:** 4
**Clarity:** The paper is well-written and clear.

**Strengths:**

The paper is well-written, with an impressively large dataset. By comparison with previous data, TwiBot22 seems significantly larger. The annotation method seems comprehensive. The dataset seems to build on the methodology and data of TwiBot20 by extending the dataset and annotation.

**Weaknesses:**

The authors perform an evaluation comparison between TwiBot and several other datasets on 35 proposed methods. While this analysis is particularly comprehensive, accuracy for TwiBot22 is not that high compared to the rest. Still it is on average over 70%, which seems fine, and it performs well for the graph-based methods, so this is not a big issue.

I appreciate the detailed Appendix, the data description together with the additional results seem consistent with the results in the main text.  However, Table 8 seems to have some issues: it seems like it's showing the average model performance (F-1) in addition to the table in the main text (which shows accuracy), but the average F-1 for TwiBot22 seems significantly worse than for the other datasets. Unless it's a reporting issue (some typo), this seems significant. For example, by Rodriguez-Ruiz, C-15 is 82.6, C-17 is 86.6, TwiBot-20 is 75.9, while TwiBot22 is 0.8. I am assuming this is a reporting issue given that TwiBot20 is not that different.

I appreciate the short discussion on societal impact and the potential of the dataset to be used to build bot evasion techniques. It would be great to have a slightly longer discussion on that and assess this potential by analyzing the related literature.

**Additional Feedback:**

Detail: missing period at the end of the line in line 599

**Correctness:**

The paper overall seems correct, with a clear explanation of the dataset construction. The evaluation methods are explicit. However, one concern I have is regarding Table 8 in the Appendix, which shows the F-1 score of several datasets (including TwiBot22) on 35 different evaluation methods. TwiBot22 has a very different score than the other datasets. I'm suspecting a typo, but this should be clarified.

**Documentation:**

The paper contains sufficient details on how the data was collected and where it is maintained. The authors do a good job in describing its usage and benchmark details.

**Ethics:**

The authors include a discussion of potential usage of this dataset in unethical ways, such as for bot evasion methods. I appreciate this acknowledgement and future work could include analyzing this potential for harm.

**Relation To Prior Work:**

The paper discusses related work well, by comparing previously collected datasets and describing methods for evaluation proposed in the field. The paper does a good job in explaining the different types of datasets used for bot detection, describing the differences between feature-based, text-based, and graph-based methods.

**Summary And Contributions:**

The paper proposes a new graph-based Twitter bot detection dataset. The authors describe the dataset in detail together with the annotation method. TwiBot22 is significantly larger than other similar datasets, with almost 1 million users. The authors use an annotation method by employing several human annotators and using this manual annotation in a weak supervised manner to annotate the rest of the dataset. The paper evaluates TwiBot22 on several methods for bot detection, comparing the accuracy of the methods with other similar datasets.

---

> ### Author Response · Authors · 2022-08-10
> **Performance comparison between TwiBot-20 and TwiBot-22. (1/3)**
>
> Thank you for your detailed and thoughtful review of our submission: *TwiBot-22: Towards Graph-Based Twitter Bot Detection*. We hereby address the raised questions and concerns. We believe the paper would be much stronger thanks to your feedback.
>
> ```
> "While this analysis is particularly comprehensive, accuracy for TwiBot22 is not that high compared to the rest. Still it is on average over 70%, which seems fine, and it performs well for the graph-based methods, so this is not a big issue."
> ```
>
> We believe that the performance drop from TwiBot-20 to TwiBot-22 could be attributed to “bot evolution” [1,2,3,4], where Twitter bots are constantly evolving to evade detection, making the performance of existing bot detectors worse over time. Since TwiBot-22 is the most recent dataset, we believe it is reasonable that model performance on TwiBot-22 is lower than TwiBot-20, which was proposed two years ago. In addition, this performance discrepancy suggests that “Twitter bot detection is still an open problem that calls for further research” (Line 257), further proving the value and importance of TwiBot-22.
>
> [1] Yang, Chao, Robert Harkreader, and Guofei Gu. "Empirical evaluation and new design for fighting evolving twitter spammers." IEEE Transactions on Information Forensics and Security 8.8 (2013): 1280-1293.
>
> [2] Cresci, Stefano. "A decade of social bot detection." Communications of the ACM 63.10 (2020): 72-83.
>
> [3] Cresci, Stefano, et al. "The paradigm-shift of social spambots: Evidence, theories, and tools for the arms race." Proceedings of the 26th international conference on world wide web companion. 2017.
>
> [4] Feng, Shangbin, et al. "BotRGCN: Twitter bot detection with relational graph convolutional networks." Proceedings of the 2021 IEEE/ACM International Conference on Advances in Social Networks Analysis and Mining. 2021.

---

> > ### Comment · Reviewer_4AAD · 2022-08-17
> > **This is helpful but wondering more**
> >
> > Thank you for the answer, I can se that the performance may be related to bot adaptation. It seems like other reviewers had a similar concern. With bot adapting to newer datasets, I agree that our methods also need to adapt and am wondering what directions for that do the authors see? Nevertheless, I get it.

---

> > > ### Author Response · Authors · 2022-08-22
> > > **Future directions for adaptable bot detection models.**
> > >
> > > Bot adaptation is indeed an important phenomenon and a major obstacle in designing bot detection models with enduring real-world impact. There were some attempts at this issue in the past [1,2,3], but personally, I feel that we need a more rigorous evaluation of an approach's ability to "adapt" to evolving bots. What data should we use? How should we design the experiments so that the results truly reflect the real-world bot evolution? There are still many open questions.
> > >
> > > Evaluation aside, we are thinking about using mixture-of-experts or continual learning for more adaptable bot detection frameworks. This is still a very preliminary idea and hopefully, we would have some results to share with the research community soon. Thank you for asking! :)
> > >
> > > [1] Feng, Shangbin, et al. "Satar: A self-supervised approach to twitter account representation learning and its application in bot detection." Proceedings of the 30th ACM International Conference on Information & Knowledge Management. 2021.
> > >
> > > [2] Cresci, Stefano, et al. "On the capability of evolved spambots to evade detection via genetic engineering." Online Social Networks and Media 9 (2019): 1-16.
> > >
> > > [3] Yang, Chao, Robert Harkreader, and Guofei Gu. "Empirical evaluation and new design for fighting evolving twitter spammers." IEEE Transactions on Information Forensics and Security 8.8 (2013): 1280-1293.

---

> ### Author Response · Authors · 2022-08-10
> **New F1-score results. (2/3)**
>
> ```
> "I appreciate the detailed Appendix, the data description together with the additional results seem consistent with the results in the main text. However, Table 8 seems to have some issues: it seems like it's showing the average model performance (F-1) in addition to the table in the main text (which shows accuracy), but the average F-1 for TwiBot22 seems significantly worse than for the other datasets. Unless it's a reporting issue (some typo), this seems significant. For example, by Rodriguez-Ruiz, C-15 is 82.6, C-17 is 86.6, TwiBot-20 is 75.9, while TwiBot22 is 0.8. I am assuming this is a reporting issue given that TwiBot20 is not that different."
> ```
>
> Thank you for your detailed reading. Minor mistakes were unfortunately made in Table 8 and we hereby correct the results. We will also update the F1-score table in the final version. Please check the following link.
>
> https://github.com/LuoUndergradXJTU/TwiBot-22#f1

---

> > ### Comment · Reviewer_4AAD · 2022-08-17
> > **This addresses my main concern**
> >
> > Thank you for the revision, this new table addresses my main concern

---

> ### Author Response · Authors · 2022-08-10
> **Expand on societal impact discussion. (3/3)**
>
> ```
> "I appreciate the short discussion on societal impact and the potential of the dataset to be used to build bot evasion techniques. It would be great to have a slightly longer discussion on that and assess this potential by analyzing the related literature."
> ```
>
> Thank you for your suggestion. We will elaborate on the potential of graph adversarial attack and TwiBot-22 as well as how we aim to prevent that. We will add this to the paper and enrich the societal impact discussion.
>
> Graph adversarial attack aims to fool GNN models by tampering with node attributes and network structure [1,2]. Recent research also used graph adversarial learning in social network analysis research [3,4]. Similarly, bot operators could exploit TwiBot-22 and techniques in graph adversarial learning to design frameworks that guide bot behavior. This type of framework could draw from the bot characteristics in TwiBot-22 to provide suggestions for where, how, and when bots should attack to achieve maximum influence. This would go against the intended use of TwiBot-22, which is facilitating the design of bot detection algorithms and helping curb the online infodemic.
>
> To this end, we plan to introduce an application process for researchers to access TwiBot-22. Firstly, one needs to contact the TwiBot-22 team via an email address that belongs to an academic or research institution. We will then ask the individual to provide relevant information such as project description, research advisor, and intended use. After that, a panel from the TwiBot-22 team would convene once a week to discuss these applications and approve them only if the applicant successfully demonstrates that their purpose is research-only. We hope that in this way bot operators and malicious actors would be prohibited from accessing TwiBot-22 and thus mitigate its potentially negative societal impact.
>
> [1] Zügner, Daniel, Amir Akbarnejad, and Stephan Günnemann. "Adversarial attacks on neural networks for graph data." Proceedings of the 24th ACM SIGKDD international conference on knowledge discovery & data mining. 2018.
>
> [2] Sun, Lichao, et al. "Adversarial attack and defense on graph data: A survey." arXiv preprint arXiv:1812.10528 (2018).
>
> [3] Yang, Xiaoyu, et al. "Rumor detection on social media with graph structured adversarial learning." Proceedings of the twenty-ninth international conference on international joint conferences on artificial intelligence. 2021.
>
> [4]Kumar, Chetan, Riazat Ryan, and Ming Shao. "Adversary for social good: Protecting familial privacy through joint adversarial attacks." Proceedings of the AAAI conference on artificial intelligence. Vol. 34. No. 07. 2020.

---

> > ### Comment · Reviewer_4AAD · 2022-08-17
> > **This is helpful but how do we handle bottlenecks in accessing data**
> >
> > Thank you to the authors for their detailed reply! I agree that some gate-keeping can be helpful in making sure that the data will not be used for malicious purposes.. yet it's also a tricky method, in ensuring that the data can be accessed well by researchers and not leaked by mistake. (e.g. it might be difficult to ensure that a committee can meet weekly after people have graduated / moved institutions as it naturally happens) I am wondering whether there are any ways to draw from adversarial learning to protect the data while keeping its usage -- one of the other reviewers mentioned privacy concerns which were addressed by the authors, is there any way to access the data through particular api calls?

---

> > > ### Author Response · Authors · 2022-08-22
> > > **Accessing the data through API calls.**
> > >
> > > Thank you for your constructive feedback! Yes, there is one straightforward approach to implementing "accessing the data through particular api calls." We could do so by only providing the ID of Twitter users and tweets while asking researchers to retrieve Twitter content with Twitter APIs by themselves. However, I could see two major issues:
> > >
> > > - Twitter bots don't keep their malicious content forever and they tend to delete malicious tweets and images after a period of time [1,2]. In addition, certain types of bots are dormant most of the time and only attack at a given time period [3,4]. As a result, when researchers re-crawl the information of these bots in the future, there might not be any malicious content left in the user's profile.
> > >
> > > - TwiBot-22 promotes a fair and comprehensive benchmarking of existing bot detection models. Since Twitter users delete old content and post new content over time, the input to bot detection models would also change under the api-call scenario, which might harm the objectiveness of the benchmarking process.
> > >
> > > I agree that people move institutions (as I will soon move too) and it might be difficult to convene a panel in the future. I'm thinking about using a system such as google forms to automatically collect TwiBot-22 access requests, have these requests automatically forwarded to all members of the team, and approve their request if 3 or more researchers say yes and no one has any concerns. Let me know if you have any suggestions about managing TwiBot-22 access.
> > >
> > > [1] Broniatowski, David A., et al. "Weaponized health communication: Twitter bots and Russian trolls amplify the vaccine debate." American journal of public health 108.10 (2018): 1378-1384.
> > >
> > > [2] Al-Rawi, Ahmed, and Vishal Shukla. "Bots as active news promoters: A digital analysis of COVID-19 tweets." Information 11.10 (2020): 461.
> > >
> > > [3] Takacs, Richard, and Ian McCulloh. "Dormant bots in social media: Twitter and the 2018 US senate election." 2019 IEEE/ACM International Conference on Advances in Social Networks Analysis and Mining (ASONAM). IEEE, 2019.
> > >
> > > [4] Rossi, Sippo, et al. "Detecting political bots on Twitter during the 2019 Finnish parliamentary election." Proceedings of the 53rd Hawaii international conference on system sciences. 2020.

---

### Review · Ethics_Reviewer_agaf · 2022-08-19

**Recommendation:** 2

**Ethics Documentation:**

There are two separate concerns that I recommend the authors address to improve the transparency of the dataset creation process:
1. Crawling software
In the paper, "TwiBot-22 conducts diversity-aware BFS starting from @NeurIPSConf". I did not check the source code, but please add into the paper (or Appendix A.2) a pointer to how to run the crawler.
2. Individual annotation information
Since the annotations were done with a pool of annotators, please ensure that the raw annotations are available in the dataset. For data with multiple annotators, the variation between annotators is a useful and interesting aspect of the data itself.

In addition, the dataset should be hosted on a platform that provides a DOI, such as zenodo. Furthermore, the license of the data should be part of the dataset itself (and not only a mention in the appendix of a paper).


**Ethics Review:**

The paper presents a dataset of scraped Twitter information, along with their relationships. Twitter information is generally considered to be public, and the community is accepting of large scale collection and use of this information.

The authors do not need to address this question, but it is important that the machine learning community discusses the issue of social media datasets. Major issues include: informed consent of the scraping of data, the almost impossible task of anonymization, and the GDPR right to be forgotten from the dataset. Ethics reviews such as this one is not a suitable approach to shift the norms of the machine learning community.

---

> ### Author Response · Authors · 2022-08-22
> **Thank you for your constructive feedback.**
>
> We address the raised concerns as follows:
>
> - The crawling software is available on the TwiBot-22 GitHub repository (https://github.com/LuoUndergradXJTU/TwiBot-22/tree/master/src/crawler). Users need to apply for Twitter API access, follow the readme, and reproduce the data collection process. We will add the sentence "The crawling software used to generate TwiBot-22 is available on the TwiBot-22 GitHub repository." to appendix A.2.
>
> - We add the individual annotation to the TwiBot-22 GitHub repository at https://github.com/LuoUndergradXJTU/TwiBot-22/blob/master/src/annotation/record_combine.csv. We agree that this information is an interesting aspect of TwiBot-22 and could be further explored.
>
> - TwiBot-22 is hosted on zenedo at https://doi.org/10.5281/zenodo.7012904. TwiBot-22 now has a clear license in the dataset itself on zenedo.
>
> Let us know if there are other things we could do to improve this work. :)

---

### Author Response · Authors · 2022-08-10
**New tables are presented as GitHub links.**

Dear reviewers, some tables could not fit into the character limit of OpenReview comments. As a result, we upload them to the TwiBot-22 GitHub repository and provide links in the comments. Thank you for your understanding.

---

### Meta-Review · Area_Chair_GP2A · 2022-09-08

**Recommendation:** Accept
**Confidence:** 5

**Metareview:**

In this work, the utilize a graph based approach to create the largest Twitter bot detection dataset, with over 100K bot users identified. The dataset is of great use for the social media mining community in many different aspects. The work was greatly improved after a very interactive and fruitful discussion period, making the contribution a lot more refined and useful.

Pros:
- Largest dataset available.
- Thorough evaluation
- Addition of most reviewer's suggestions have strengthen the details in the paper

Cons:
- Minor statistical rigor elements are missing, but not completely necessary.

---

### Decision · Program_Chairs · 2022-09-16

Accept